# Adaptive Quantization in Generative Flow Networks for Probabilistic Sequential Prediction

**Nadhir Hassen**[*]
University of Adelaide,
AIML and Mila – Quebec AI Institute

**Zhen Zhang**
University of Adelaide, AIML

**Johan Verjans**
University of Adelaide, AIML

## Abstract

Probabilistic time series forecasting, essential in domains like healthcare and neuroscience, requires models capable of capturing uncertainty and intricate temporal dependencies. While deep learning has advanced forecasting, generating calibrated probability distributions over continuous future values remains challenging. We introduce Temporal Generative Flow Networks (Temporal GFNs), adapting Generative Flow Networks (GFNs) – a powerful framework for generating compositional objects – to this sequential prediction task. GFNs learn policies to construct objects (eg. forecast trajectories) step-by-step, sampling final objects proportionally to a reward signal. However, applying GFNs directly to continuous time series necessitates addressing their inherently discrete action spaces and ensuring differentiability. Our framework tackles this by representing time series segments as states and sequentially generating future values via quantized actions chosen by a forward policy. We introduce two key innovations: (1) An adaptive, curriculum-based quantization strategy that dynamically adjusts the number of discretization bins based on reward improvement and policy entropy, balancing precision and exploration throughout training. (2) A straight-through estimator mechanism enabling the forward policy to output both discrete (hard) samples for trajectory construction and continuous (soft) samples for stable gradient propagation. Training utilizes a trajectory balance loss objective, ensuring flow consistency, augmented by an entropy regularizer. We provide rigorous theoretical bounds on the quantization error's impact and the adaptive factor's range. We demonstrate how Temporal GFNs offer a principled way to leverage the structured generation capabilities of GFNs for probabilistic forecasting in continuous domains.

## 1 Introduction

Time series forecasting – predicting future values based on historical observations – is fundamental to informed decision-making across countless applications, from managing energy grids and financial portfolios to monitoring patient health and understanding brain activity [14, 7]. In many critical domains, particularly healthcare (e.g., forecasting vital signs from Electronic Health Records - EHRs) and neuroscience (e.g., predicting Electroencephalogram - EEG signals), obtaining not just a single point prediction but a full *probabilistic* forecast is paramount. Probabilistic forecasts quantify the inherent uncertainty in predictions, providing prediction intervals or complete density estimations, crucial for risk assessment and reliable decision support [10]. The advent of deep learning has

---

[*]Correspondence to `nadhir.hasseng@mila.quebec`, `nadhir.hassen@adelaide.edu.au`

39th Conference on Neural Information Processing Systems (NeurIPS 2025).

revolutionized time series forecasting. Models based on Recurrent Neural Networks (RNNs) [30, 27] and Transformers [21, 25, 32] can capture complex temporal patterns from large datasets. These models often achieve probabilistic forecasts by parameterizing a specific output distribution (e.g., Gaussian [30]) or predicting specific quantiles [34]. While effective, these approaches can impose potentially restrictive assumptions on the shape of the predictive distribution or suffer from issues like quantile crossing. More flexible generative models like Normalizing Flows [28] and Diffusion Models [19] have also been adapted, offering richer distributional modeling at potentially higher computational costs. Recently, Large Language Models (LLMs) have demonstrated remarkable zero-shot capabilities in various domains, including time series forecasting [12]. Frameworks like Chronos [1] show that standard transformer architectures, trained on time series tokenized via simple scaling and quantization, can achieve state-of-the-art results. This highlights the power of sequence modeling and the potential of mapping continuous time series to a discrete vocabulary. However, these methods typically rely on a *fixed* quantization scheme decided a priori. An alternative paradigm for structured generation is offered by Generative Flow Networks (GFNs) [3, 6]. GFNs learn policies to construct complex objects (like molecules or computational graphs) through a sequence of discrete actions in a state space. The key idea is to sample terminal objects (completed structures) with a probability proportional to a given reward function, effectively learning to navigate the state space towards high-reward configurations. This focus on sampling from a reward-modulated distribution makes GFNs naturally suited for tasks where we want to generate diverse, high-quality samples, rather than just finding a single optimal solution (as in reinforcement learning). This generative process seems well-aligned with time series forecasting: we can view a forecast trajectory as a compositional object built step-by-step, where the *reward* reflects the accuracy of the final forecast. However, GFNs traditionally operate in discrete state and action spaces. How can we adapt them to generate sequences of *continuous* time series values? Directly applying GFNs faces two primary hurdles: 1. **Continuous Action Space:** Time series values are continuous, but GFN policies typically output probabilities over discrete actions. 2. **Differentiability:** Selecting a discrete action (even if representing a continuous value via quantization) breaks the gradient flow required for training neural network policies.

In this work, we propose **Temporal Generative Flow Networks (Temporal GFNs)**, a framework designed to overcome these challenges and harness GFNs for probabilistic continuous time series forecasting. Our approach leverages the strengths of GFNs – structured sequential generation and reward-driven sampling – while introducing mechanisms specifically tailored for the continuous nature of time series data. The main framework is described in figure 1.

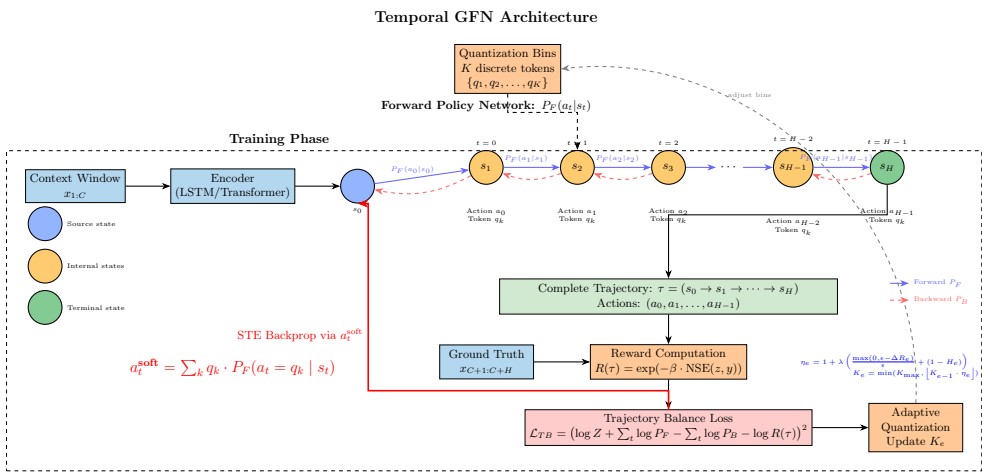

Figure 1: Temporal GFN framework: the encoder conditions on the context window; a forward policy samples actions via quantized bins to form the trajectory; a reward module scores it; Trajectory Balance drives learning; STE provides differentiability through discrete steps; and an adaptive K loop adjusts resolution as training stabilizes.

Our core contributions are:

1. **GFN Formulation for Probabilistic Forecasting:** We frame forecasting as constructing a trajectory in a GFN state space, where states are time series windows and actions append the next predicted (quantized) value. The GFN learns to sample trajectories whose associated rewards (forecast accuracy) match the target distribution.

2. **Adaptive Curriculum-Based Quantization:** We move beyond fixed quantization by introducing a dynamic scheme. The number of quantization bins $K$ is adjusted during training based on reward improvement and policy entropy, enabling the model to learn effectively with adaptive precision.

3. **Differentiable Discrete Actions via STE:** We employ a straight-through estimator (STE) [5] to reconcile discrete action selection with gradient-based optimization. The forward policy generates both discrete samples for state transitions and continuous approximations for gradient propagation.

4. **Trajectory Balance Learning:** We train the framework using the Trajectory Balance (TB) loss [23], enforcing flow consistency across entire forecast trajectories, and include an entropy bonus to encourage exploration.

5. **Theoretical Guarantees:** We provide theoretical bounds on the policy gradient error introduced by our quantization and STE approach, and on the range of the adaptive quantization factor, ensuring controlled behavior.

The paper is structured as follows: Section 2 reviews related work in forecasting and Generative Flow Networks (GFNs). Section 3 introduces the Temporal GFN framework, detailing its components like adaptive quantization, STE, and the training objective. Section 4 provides empirical validation through mechanism-focused ablation studies and performance comparisons against benchmarks. Section 5 discusses the implications of our approach, and 6 concludes with a summary and future directions.

## 2 Background and Related Work

Our work builds upon concepts from probabilistic time series forecasting, Generative Flow Networks, and methods for handling continuous data in discrete frameworks.

### 2.1 Time Series Forecasting

Time series forecasting aims to predict future values $x_{C+1:C+H} = [x_{C+1}, \ldots, x_{C+H}]$ given historical context $x_{1:C} = [x_1, \ldots, x_C]$. We focus on *probabilistic* forecasting, which involves predicting the conditional probability distribution $p(x_{C+1:C+H}|x_{1:C})$. **Classical methods** like ARIMA and ETS [16] model time series components (trend, seasonality, noise) and often provide prediction intervals under specific statistical assumptions. **Deep learning methods** leverage architectures like RNNs [30, 27] or Transformers [21, 25] to learn complex dependencies. Probabilistic outputs are commonly achieved by parameterizing output distributions [30], predicting quantiles [34], or using generative models like Flows [28] or Diffusion Models [19]. **LLM-based forecasters** [12, 1] have shown strong performance, often by tokenizing time series values into a discrete vocabulary similar to natural language. Chronos [1], for instance, uses simple scaling and uniform quantization, demonstrating the effectiveness of applying standard language model architectures and training objectives (cross-entropy) to tokenized time series. While powerful, these often rely on fixed quantization and may require very large models. Our Temporal GFN differs by using a GFN objective and introducing adaptive quantization.

### 2.2 Generative Flow Networks (GFNs)

GFNs [3, 6] are a class of probabilistic generative models designed to learn policies for constructing objects $x$ in a compositional manner, such that the probability of sampling $x$ is proportional to a given reward $R(x) > 0$. They operate on a directed acyclic graph (DAG) where nodes represent states $s$ (partial or complete objects) and edges represent actions $a$ leading from one state to another. GFNs excel at exploring complex, high-dimensional discrete spaces and sampling diverse, high-reward objects [17, 36]. Applying them to continuous domains, especially for sequential generation, remains an open challenge [20].

## 2.3 Discretization and Continuous Actions in Time Series Forecasting

Bridging continuous time series data with models often designed for discrete inputs is a central challenge in modern forecasting. **Quantization**, or tokenization, which maps continuous values to a finite set of discrete symbols, is a widely adopted strategy. This has enabled the successful application of powerful sequence models, like Transformers, to time series by treating them as "sentences" of tokens [1] or like amino acids in proteins navigating a continuous space to model discrete sequences [2]. Recent advancements also explore more sophisticated tokenization schemes, such as wavelet-based approaches, to better capture multi-scale temporal features before feeding them to foundation models [24]. While these methods highlight the power of discretization, a commonality is often the reliance on a *fixed* quantization scheme (e.g., a predetermined number of bins or a fixed tokenization vocabulary). As discussed by Rabanser et al. [26], such fixed schemes can be suboptimal: too few bins (coarse quantization) limit precision and lead to information loss, whereas too many bins (fine quantization) can create an overly sparse action space, hindering exploration and slowing down the learning process, particularly in early training phases. Our proposed adaptive quantization mechanism (Section 3) is designed to directly mitigate this by dynamically adjusting the quantization granularity. Furthermore, when discrete actions (like selecting a quantization bin) are part of the model's generative process, as in our Temporal GFN framework, ensuring differentiability for gradient-based training of neural policies is critical. The **Straight-Through Estimator (STE)** [5, 35] serves as a standard technique to enable gradient propagation through such discrete steps. We employ STE to allow our GFN's forward policy to make discrete action selections (choosing quantized values) while facilitating effective gradient-based learning of the policy network. Our work, therefore, uniquely combines the GFN paradigm with both adaptive quantization for optimized precision and STE for differentiability, specifically targeting the challenges of probabilistic continuous time series forecasting.

## 3 Methodology: Temporal Generative Flow Networks

In this section, we detail the proposed Temporal Generative Flow Network framework. We begin by mapping the probabilistic time series forecasting problem onto the Generative Flow Network paradigm. Subsequently, we describe the core components designed to handle continuous data within this framework: state representation, the adaptive quantization scheme coupled with the forward policy network, the mechanism for ensuring differentiability, and finally, the training objective based on Trajectory Balance.

### 3.1 Forecasting as Trajectory Generation in a GFN

Generative Flow Networks (GFNs) provide a powerful lens for learning to generate complex, structured objects [6]. They learn a policy to navigate a state space through a sequence of actions, ultimately sampling terminal objects (complete structures) with probabilities proportional to a predefined reward signal. This compositional generation process, guided by reward, makes GFNs an appealing candidate for sequential prediction tasks like time series forecasting.

We conceptualize the generation of a forecast as constructing a trajectory $\tau$ in a GFN state space.

- **States** ($s$): A state $s_t$ encapsulates the information available at step $t$ of the forecast generation. We choose to represent states as fixed-length sliding windows containing the most recent history and any forecast values generated so far. The process begins from an initial state $s_0$, which corresponds to the observed historical context window, $s_0 = (x_{\text{obs}}, \ldots, x_{\text{obs}+T-1})$.

- **Actions** ($a$): An action $a_t$ taken from state $s_t$ corresponds to selecting the predicted value for the next time step, $x_{\text{obs}+T+t}$. As GFNs fundamentally operate over discrete action spaces, a crucial step is mapping the inherently continuous space of possible time series values to a finite set of discrete actions. This is achieved through quantization, detailed in Section 3.2.

- **State Transitions:** Applying a chosen (discrete) action $a_t^{\text{hard}}$ to state $s_t$ results in the next state $s_{t+1}$. This is typically achieved by appending $a_t^{\text{hard}}$ to the window represented by $s_t$ and removing the oldest value, effectively sliding the window forward: $s_{t+1} = (s_t[1], \ldots, s_t[T-1], a_t^{\text{hard}})$.

- **Trajectories ($\tau$):** A complete forecast trajectory $\tau = (s_0, s_1, \ldots, s_{T'})$ is formed by sequentially applying actions $a_0, \ldots, a_{T'-1}$ according to a learned policy, spanning the desired forecast horizon $T'$.

- **Reward ($R(\tau)$):** A reward function assigns a scalar value to each completed trajectory, quantifying the desirability or quality of the generated forecast (e.g., based on its accuracy against the true future values). The GFN's objective is to learn a policy that samples trajectories $\tau$ such that the sampling probability $P(\tau) \propto \exp(R(\tau))$.

The main technical challenges lie in adapting the discrete action mechanism of GFNs to the continuous nature of time series values and ensuring the entire process remains differentiable for effective training using gradient-based optimization.

## 3.2 Forward Policy, Adaptive Quantization, and Differentiability

The core generative component is the forward policy $P_F(a_t|s_t)$, which learns the probability distribution over possible next actions given the current state. To manage the continuous action space and enable learning, we combine quantization, a Transformer-based policy network, an adaptive mechanism for quantization granularity, and the Straight-Through Estimator (STE).

### 3.2.1 Quantization and the Forward Policy Network

To bridge the gap between continuous time series values and the GFN's discrete action space, we employ quantization. We map continuous values within a range $[v_{\min}, v_{\max}]$ to a set of $K$ discrete bins, represented by their centers:

$$q_k = v_{\min} + \frac{k-1}{K-1}(v_{\max} - v_{\min}), \quad k = 1, \ldots, K. \tag{1}$$

These $K$ bin centers $\{q_1, \ldots, q_K\}$ constitute the discrete action space $\mathcal{A}$ available to the GFN at any given state $s_t$. The forward policy network $P_F(a_t = q_k|s_t)$ computes the probability of selecting action $q_k$ given the current state window $s_t$. Recognizing the success of Transformer architectures [32] in capturing long-range dependencies in sequential data, we use a Transformer encoder as the backbone of our policy network. The state window $s_t$, a sequence of $T$ real values, is processed by the Transformer encoder, yielding a fixed-dimensional representation $h_t$ that summarizes the relevant historical context. This representation is then projected by a linear output layer to produce logits over the $K$ possible discrete actions: $\text{logits}_t = W_F h_t + b_F$, where $\text{logits}_t \in \mathbb{R}^K$. Applying the softmax function to these logits yields the policy distribution:

$$P_F(a_t = q_k|s_t) = \frac{\exp(\text{logits}_{t,k})}{\sum_{j=1}^{K} \exp(\text{logits}_{t,j})}. \tag{2}$$

This network learns to predict the probability of the *next* value falling into each of the $K$ quantization bins, based on the preceding time series pattern encoded in $s_t$.

### 3.2.2 Adaptive Quantization: A Curriculum for Precision

Using a fixed number of bins $K$ throughout training presents a dilemma common in discretization tasks [26]. A small $K$ leads to high quantization error, limiting forecast precision, while a large $K$ creates a vast, sparse action space that can significantly slow down learning and hinder exploration, especially in the initial phases when the policy is still poorly defined. To overcome this, we introduce an adaptive quantization scheme that treats the number of bins $K$ as a parameter that evolves during training, akin to a curriculum learning strategy [4]. The model starts with a relatively small $K$ (coarse granularity), making the initial exploration task simpler. As training progresses, $K$ is dynamically adjusted based on the model's performance and confidence. Specifically, after a warmup period $E_{\text{warmup}}$, we compute an adaptive update factor $\eta_e$ at each epoch $e$. This factor depends on the recent reward improvement $\Delta R_e = R_e - R_{e-\delta}$ (average reward difference over $\delta$ epochs) and the average normalized forward policy entropy $H_e \in [0, 1]$:

$$\eta_e = 1 + \lambda \left( \underbrace{\frac{\max(0, \epsilon - \Delta R_e)}{\epsilon}}_{\text{Improvement Signal}} + \underbrace{(1 - H_e)}_{\text{Confidence Signal}} \right). \tag{3}$$

Here, $\lambda > 0$ controls the sensitivity of adaptation, and $\epsilon > 0$ is a target reward improvement threshold.

- The *Improvement Signal* becomes positive if the model's reward gain is below the threshold $\epsilon$. Slow improvement suggests the current quantization might be too coarse, thus pushing $\eta_e > 1$ to encourage increasing $K$.

- The *Confidence Signal* is large when entropy $H_e$ is low (i.e., the policy is highly confident or "peaked"). High confidence might indicate premature convergence or insufficient exploration. Increasing $K$ in this case refines the action space, potentially revealing finer structures and encouraging further exploration.

The number of bins for the next epoch $K_e$ is updated multiplicatively, bounded by a maximum $K_{\max}$:

$$K_e = \min(K_{\max}, \lfloor K_{e-1} \cdot \eta_e \rfloor). \tag{4}$$

If $K_e$ changes value, the size of the final linear layer $(W_F, b_F)$ in the forward policy network (and the backward network, if learned) must be adjusted to match the new number of output bins.

To ensure training stability when the number of bins $K_e$ is increased, a simple random re-initialization of the output layer would be catastrophic, erasing previously learned knowledge. Instead, we employ a weight-reuse strategy. The weights and biases in the linear layer corresponding to the pre-existing bins are preserved. The weights and biases for the newly added bins are initialized to near-zero values. This method allows the policy to retain its learned distribution over the existing action space while cautiously exploring the new, finer-grained actions, thus preventing catastrophic forgetting and ensuring a smooth curriculum.

### 3.2.3 Ensuring Differentiability: The Straight-Through Estimator

A critical issue arises when selecting the discrete action $a_t^{\text{hard}} = q_k$ from the policy $P_F(\cdot|s_t)$ to update the state. Operations like sampling or taking the argmax are non-differentiable, which would prevent gradients from flowing back from the final loss calculation to the parameters of the policy network ($W_F, b_F$ and the Transformer encoder). To circumvent this, we employ the Straight-Through Estimator (STE) technique [5, 35]. The core idea is to use the discrete value in the forward pass for computations that require it (like state transitions) but to use a continuous approximation for the backward pass (gradient computation). Our forward policy generates two outputs at each step $t$:

- **Hard Sample ($a_t^{\text{hard}}$):** The discrete quantized value selected for constructing the trajectory. This is typically the most likely action: $a_t^{\text{hard}} = q_{\arg\max_k P_F(a_t = q_k | s_t)}$. This $a_t^{\text{hard}}$ value is used to update the state $s_t \rightarrow s_{t+1}$.

- **Soft Sample ($a_t^{\text{soft}}$):** A continuous, differentiable proxy for the chosen action, computed as the expectation of the action under the current policy distribution: $a_t^{\text{soft}} = \sum_{k=1}^{K} q_k P_F(a_t = q_k | s_t)$. During backpropagation, the gradient is computed with respect to $a_t^{\text{soft}}$. This gradient is then passed *straight through* the non-differentiable argmax/sampling operation that produced $a_t^{\text{hard}}$, effectively using the gradient of the continuous expectation $a_t^{\text{soft}}$ to update the parameters that generated the policy $P_F(\cdot|s_t)$.

This STE mechanism allows us to maintain the integrity of the discrete state transitions inherent to the GFN formulation while enabling end-to-end training with gradient descent.

### 3.3 Learning Objective: Trajectory Balance and Exploration

GFNs are trained by enforcing flow consistency conditions. We adopt the Trajectory Balance (TB) loss [23], which offers robust credit assignment, particularly for long sequences.

### 3.3.1 Backward Policy

The TB objective necessitates a backward policy $P_B(s_{t-1}|s_t)$, modeling the probability of transitioning *backward* from state $s_t$ to a potential predecessor state $s_{t-1}$. We consider two implementations:

1. **Uniform Policy:** Assumes all valid predecessor states are equally likely. Given the fixed action space size $K$, this simplifies to $\log P_B(s_{t-1}|s_t) = -\log K$. This is parameter-free but ignores temporal structure.

2. **Learned Policy:** Parameterizes $P_B$ using a neural network that takes the representations of the involved states $(h_{t-1}, h_t)$ as input, e.g., $\log P_B(s_{t-1}|s_t) = \log \text{softmax}(W_B \text{concat}(h_{t-1}, h_t) + b_B)$. This allows $P_B$ to adapt but increases model size.

### 3.3.2 Reward Function

The reward function $R(\tau)$ guides the GFN towards generating desirable trajectories. For forecasting, desirability equates to accuracy. We define the reward based on the discrepancy between the generated forecast sequence $z = (z_1, \ldots, z_{T'})$ (derived from the sequence of soft samples $a_t^{\text{soft}}$) and the ground truth future sequence $y = (y_1, \ldots, y_{T'})$. We use an exponential function of the negative normalized Mean Squared Error (MSE):

$$R(\tau) = \exp\left( -\beta \frac{1}{T'} \sum_{t=1}^{T'} \frac{(z_t - y_t)^2}{(v_{\max} - v_{\min})^2} \right), \tag{5}$$

where $\beta > 0$ controls the reward scaling. Higher values of $\beta$ create a sharper reward landscape, more strongly penalizing deviations from the ground truth.

### 3.3.3 Trajectory Balance Loss with Entropy Regularization

The TB loss [23] equates the flow along a trajectory based on forward probabilities and the initial flow $Z$ (a learnable parameter representing the partition function or total flow) with the flow based on backward probabilities and the terminal reward $R(\tau)$. The squared difference forms the loss for a single trajectory:

$$\mathcal{L}_{\text{TB}}(\tau) = \left( \log Z + \underbrace{\sum_{t=0}^{T'-1} \log P_F(s_{t+1}|s_t)}_{\text{Forward Log-Prob}} - \underbrace{\sum_{t=1}^{T'} \log P_B(s_{t-1}|s_t)}_{\text{Backward Log-Prob}} - \underbrace{\log R(\tau)}_{\text{Log Reward}} \right)^2. \tag{6}$$

The overall TB loss is the expectation over trajectories sampled from the forward policy: $\mathcal{L}_{\text{TB}} = \mathbb{E}_{\tau \sim P_F}[\mathcal{L}_{\text{TB}}(\tau)]$.

To mitigate the risk of the policy becoming prematurely deterministic and to encourage broader exploration of the action space, we incorporate an entropy bonus into the learning objective (equivalent to subtracting an entropy penalty from the loss). The entropy of the forward policy at state $s_t$ is: $\mathcal{H}(P_F(\cdot|s_t)) = -\sum_{k=1}^{K} P_F(a_t = q_k|s_t) \log P_F(a_t = q_k|s_t)$. The final loss function balances the TB objective with the average entropy across the trajectory: $\mathcal{L} = \mathcal{L}_{\text{TB}} - \lambda_{\text{entropy}} \mathbb{E}_{\tau \sim P_F} \left[ \frac{1}{T'} \sum_{t=0}^{T'-1} \mathcal{H}(P_F(\cdot|s_t)) \right]$, where $\lambda_{\text{entropy}} \geq 0$ is a hyperparameter controlling the weight of the entropy regularization. Optimizing this final loss $\mathcal{L}$ trains the forward policy $P_F$, the backward policy $P_B$ (if learned), and the partition function $Z$, we summarize this section with Algorithm A and we provide theorical analysis of our methodology in Appendix B.

## 4  Experiments

In this section, we present a comprehensive empirical evaluation of the Temporal Generative Flow Network (Temporal GFN) framework. Our experimental investigation is twofold. First (Section 4.1), we assess the performance of optimized Temporal GFN configurations against a broad range of established baselines on standard forecasting benchmarks and challenging related datasets. Throughout this section, we explicitly link empirical observations to the mathematical concepts introduced in Appendix B. Detailed experimental setup common to both studies is described in Appendix D. Second (Section 4.2), we conduct mechanism-focused analyses and ablations using synthetic data and standard benchmarks to validate the theoretical underpinnings of our approach, particularly concerning adaptive quantization and the Straight-Through Estimator (STE).

## 4.1 Benchmark Performance Comparison

**Datasets, Training Strategy and Evaluation Metrics.** We train and evaluate Temporal GFN on a large and diverse collection of publicly available time series datasets, comprehensively gathered and curated, mirroring the scale used in recent foundation model studies [1], the dataset, the training strategy, evaluation tasks and metrics are detailed in Appendix I. We evaluate optimized Temporal GFN configurations against relevant baselines across the defined benchmarks.

### 4.1.1 Comparison with RL and MCMC Baselines (Benchmark I)

Table 1 summarizes the aggregated relative performance on Benchmark I against PPO, SAC, and MCMC baselines. Temporal GFN (Adaptive K20) shows substantial improvements (25-35%) across all metrics compared to these methods. The Learned Policy variant further improves probabilistic metrics (CRPS/WQL). This highlights the benefits of the GFN framework tailored for forecasting over generic sequential decision-making approaches shown in table 1.

Table 1: Aggregated Relative Performance vs. RL/MCMC on Benchmark I.

| Method | CRPS ($\downarrow$) | WQL ($\downarrow$) | MASE ($\downarrow$) | Avg. Improv. (%) |
|---|---|---|---|---|
| Temporal GFN (Adaptive K20) | 0.1674 | 0.2273 | **0.8938** | - |
| Temporal GFN (Fixed K20) | 0.1781 | 0.2401 | 0.9363 | -5.9% |
| Temporal GFN (Learned Pol. Adapt K10) | **0.1453** | **0.2137** | 1.0345 | +6.0%* |
| PPO | 0.2299 | 0.3013 | 1.1759 | -25.1% |
| SAC | 0.2423 | 0.3227 | 1.2491 | -29.5% |
| MCMC | 0.2673 | 0.3522 | 1.3523 | -35.5% |

*Avg. Improv. for Learned Policy calculated based on CRPS/WQL vs Adaptive K20.

### 4.1.2 Comparison with SOTA Time Series Forecasters (Benchmark I & II)

Table 2 presents aggregated performance against SOTA deep learning models, evaluated on both Benchmark I (In-Domain) and Benchmark II (Zero-Shot) where applicable. Temporal GFN demonstrates highly competitive performance, often leading in probabilistic metrics (CRPS, WQL) and particularly excelling in zero-shot scenarios and on metrics sensitive to distribution shape like Calibration Error and Multimodality Score (evaluated on extended/healthcare benchmarks, details in Appendix H). Qualitative examples in Appendix J (Figures 21) visually support Temporal GFN's ability to capture complex dynamics and multimodality.

Table 2: Aggregated Relative Performance vs. SOTA Models (In-Domain / Zero-Shot).

| Model | Eval Setting | CRPS ($\downarrow$) | WQL ($\downarrow$) | MASE ($\downarrow$) | Calib. Err. ($\downarrow$) | Multimod. ($\uparrow$) |
|---|---|---|---|---|---|---|
| Temporal GFN | Zero-Shot / In-Domain | **0.1542** | **0.2158** | **0.9378** | **0.0348** | **0.8762** |
| Lag-Llama | Zero-Shot / Pretrained | 0.1675 | 0.2401 | 0.9532 | 0.0623 | 0.7234 |
| Chronos (Base) | Zero-Shot / Pretrained | 0.1731 | 0.2534 | 1.0273 | 0.0571 | 0.6957 |
| MOREI (Base) | Task-Specific Trained | 0.1859 | 0.2678 | 1.0871 | 0.0745 | 0.6182 |
| TimeFM | Task-Specific Trained | 0.2076 | 0.2846 | 1.1258 | 0.1183 | 0.3012 |
| Temporal GFN (Learned Pol.) | In-Domain | 0.1453 | 0.2137 | 1.0345 | - | - |

Main Temporal GFN results reflect strong performance across settings. Calib. Err. & Multimod. primarily from healthcare/extended benchmarks. Task-Specific models trained per dataset. Learned Policy results from Table 3.

Table 3: Ablation Study Configurations and Main Results.

| Experiment Config | Quantization | Start K | Policy Type | CRPS | WQL | MASE |
|---|---|---|---|---|---|---|
| Fixed K=10 | fixed | 10 | uniform | 0.1546 | 0.2346 | 1.1946 |
| Fixed K=20 | fixed | 20 | uniform | 0.1921 | 0.2521 | 0.9721 |
| Adaptive K=10 | adaptive | 10 | uniform | 0.1688 | 0.2408 | 1.1048 |
| Adaptive K=20 | adaptive | 20 | uniform | 0.1845 | 0.2385 | **0.9532** |
| Learned Policy (Adapt K10) | adaptive | 10 | learned | **0.1453** | **0.2137** | 1.0345 |

## 4.2 Analysis of Quantization Mechanism

We investigate the quantitative relationship between quantization resolution (number of bins, $K$), the resulting approximation error, and the implications for policy learning and performance. Table 3 summarizes the configurations and main results for this ablation study.

**Quantization Error Bound Validation:** Theorem B.1 established that the policy gradient error depends linearly on the quantization error $\epsilon$. We empirically verify the relationship between $K$ and $\epsilon$. Figure 2 (bottom left) plots the Mean Squared Error (MSE), serving as a proxy for $\epsilon^2$, against $K$. As predicted, increasing $K$ significantly reduces the quantization error, consistent with the premise that finer discretization leads to smaller $\epsilon$. The relationship between error reduction and reward gain is explored in Figure 3. The left panel shows a strong negative correlation between quantization error and achievable reward, while the right panel shows that increasing $K$ yields diminishing returns in reward improvement, suggesting a trade-off that motivates adaptive approaches.. We visually confirms this reduction in approximation error across the data distribution for increasing $K$ in Appendix J Figure 4.

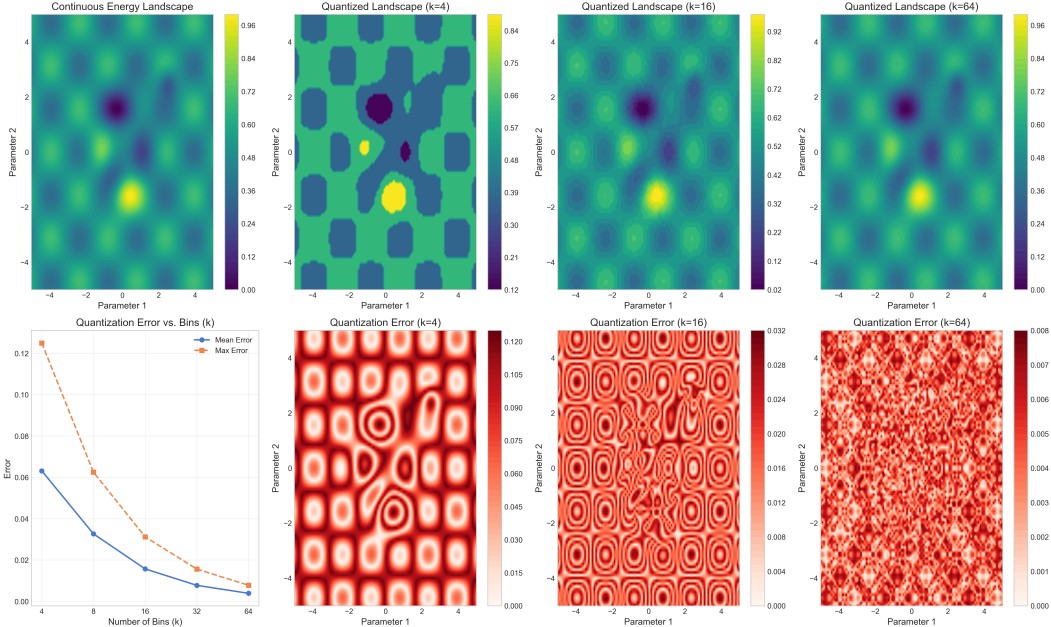

Figure 2: Gradient flow fields: Continuous (left) vs. Quantized with STE (right, K=16), validating STE.

**Adaptive Quantization Dynamics and Bounded Updates:** The adaptive mechanism (Section 3.2.2) aims to optimize $K$ based on reward improvement ($\Delta R_e$) and policy entropy ($H_e$), as governed by Eq. 3. Theorem **??** guarantees the update factor $\eta_e$ is bounded: $1 \leq \eta_e \leq 1 + 2\lambda$. This prevents uncontrolled explosion or collapse of $K$. In Appendix J, figure 11 (left panel, purple/blue lines for Adaptive Quantization) shows the coupled evolution of reward and entropy, which drive $\eta_e$. The resulting non-uniform bin distribution learned by adaptive quantization (Figure 5, Adaptive panel) demonstrates its ability to allocate bins effectively compared to fixed uniform quantization (Uniform panel). This optimized allocation minimizes the effective $\epsilon$ in relevant regions, leading to superior performance observed in benchmark comparisons (Tables 1, 3).

**Quantization and Policy Representation:** In Figure 6 Appendix J visualizes how the choice of $K$ impacts the policy's representation of the underlying continuous energy landscape. A low $K$ provides a very coarse approximation, while higher $K$ values capture more detail but remain discrete. The adaptive mechanism, by optimizing bin placement, aims to create a quantized landscape that best preserves the essential features (modes, relative probabilities) of the continuous landscape within the constraints of $K$ bins, facilitating more accurate policy learning.

**Entropy-Error Trade-off Quantification:** In Figure 7 Appendix J quantifies the trade-off inherent in selecting $K$. It plots Normalized MSE (proxy for quantization error $\epsilon$) against Normalized Entropy (uniformity of bin usage). As K increases (indicated by color/size), error decreases, but entropy might slightly decrease from its maximum if adaptation leads to concentrated bins. This plot empirically illustrates the optimization landscape navigated by the adaptive mechanism.

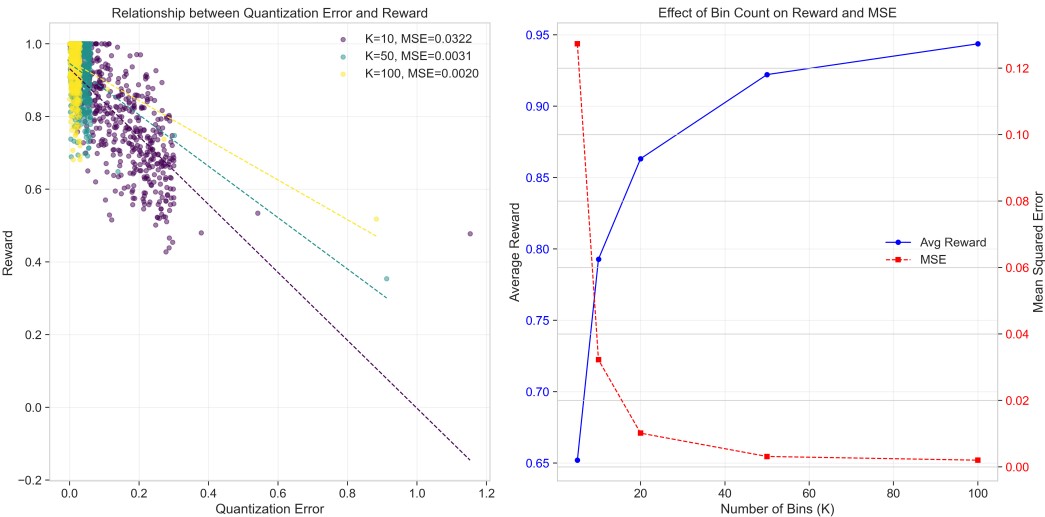

Figure 3: Reward and Entropy dynamics over training and their relationship for different configurations.

## 5   Discussion

While our results demonstrate the effectiveness of Temporal GFNs, particularly in probabilistic accuracy and multimodality capture, it is important to contextualize its limitations. The sequential, step-by-step generation of trajectories, fundamental to the GFN paradigm, can be more computationally intensive during training compared to models like standard Transformers or LLMs that generate forecasts in a single forward pass. Furthermore, the performance is contingent on a well-designed reward function $R(\tau)$. Although the exponential MSE reward used in our work is effective for accuracy, designing rewards for more complex or domain-specific objectives (e.g., penalizing physiologically implausible trajectories in healthcare) remains a critical and potentially challenging modeling decision.

**Adaptive quantization.**    Our methodology (Section 3) robustly applies GFNs to continuous data by dynamically balancing precision and exploration. Governed by reward and entropy signals (Eq. 3) and stabilized by theoretical bounds (Theorem B.2 Appendix B), it acts as an effective learning curriculum. While alternative *data-driven quantization* strategies (e.g., quantile-based bins) were explored in [9], particularly for skewed data by reducing quantization error $\epsilon$ more aggressively in dense regions (Figure 9, leading to CRPS improvements in Figure 10), it introduces significant implementation complexity (e.g., estimating quantiles dynamically or learning boundaries) and potentially higher training overhead (15% reported increase). Our adaptive uniform spacing approach provides a strong balance, achieving competitive performance (Tables 1, 2) and adapting the *number* of bins effectively based on learning dynamics (Figure 11) without the full complexity of learning non-uniform boundaries, representing a practical and performant design choice. The reduction in $\epsilon$ achieved through adaptation directly contributes to improved gradient accuracy (Theorem C.1, Eq. 9).

**Gradient flow.**    For enabling gradient flow through discrete quantization choices, we employ the *Straight-Through Estimator (STE)* [5]. While STE is a widely adopted and practically effective method, alternatives like *Gumbel-Softmax (GS)* relaxation [18] offer a different approach to managing

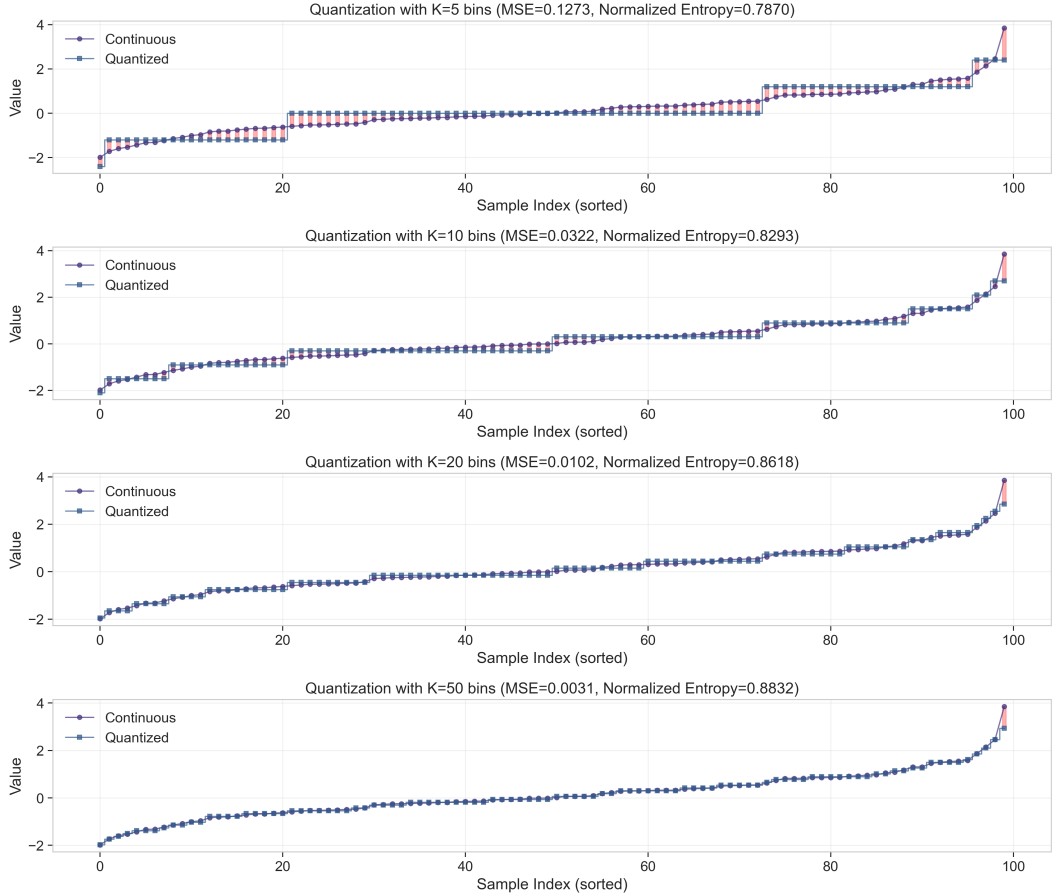

Figure 4: Quantization of sorted continuous data points. Visualizes the reduction in MSE (red area) as K increases from 5 to 50.

differentiability in discrete action spaces via $z_k = \text{softmax}((\ell_k + g_k)/\tau_{temp})$. Our comparative analysis highlights the nuanced trade-offs. Figure 12 shows that GS can lead to faster initial convergence in terms of training loss, potentially due to its inherently smoother gradient signals early in training (high $\tau_{temp}$). However, this initial advantage can sometimes plateau, with STE achieving a slightly lower final loss in some configurations. A key factor is gradient stability; Figure 8 illustrates that while STE maintains relatively stable gradient variance throughout training, GS, despite generally lower average variance, can exhibit occasional spikes, particularly if temperature annealing is not perfectly tuned. Furthermore, as detailed in Figure 13, GS introduces notable overhead: approximately 35% increased compute time, 41% higher memory usage, and a significant 70% rise in perceived implementation complexity due to temperature annealing and managing the continuous relaxation. Considering these factors, Figure 14 summarizes our practical guidance: STE is preferred for its simplicity, resource efficiency, and stability, especially for smaller models or complex distributions where its biased but consistent gradient proves effective.

**Entropy Regularization.** The combination of the *Trajectory Balance* objective with *entropy regularization* ($\mathcal{L}$) proves highly effective. TB loss facilitates credit assignment, while the entropy bonus explicitly manages the exploration-exploitation trade-off, crucial for learning diverse and well-calibrated distributions. Removing the entropy bonus leads to significantly poorer final rewards (Figure 15), highlighting its role in preventing mode collapse and ensuring sufficient exploration (Figures 16, 17).

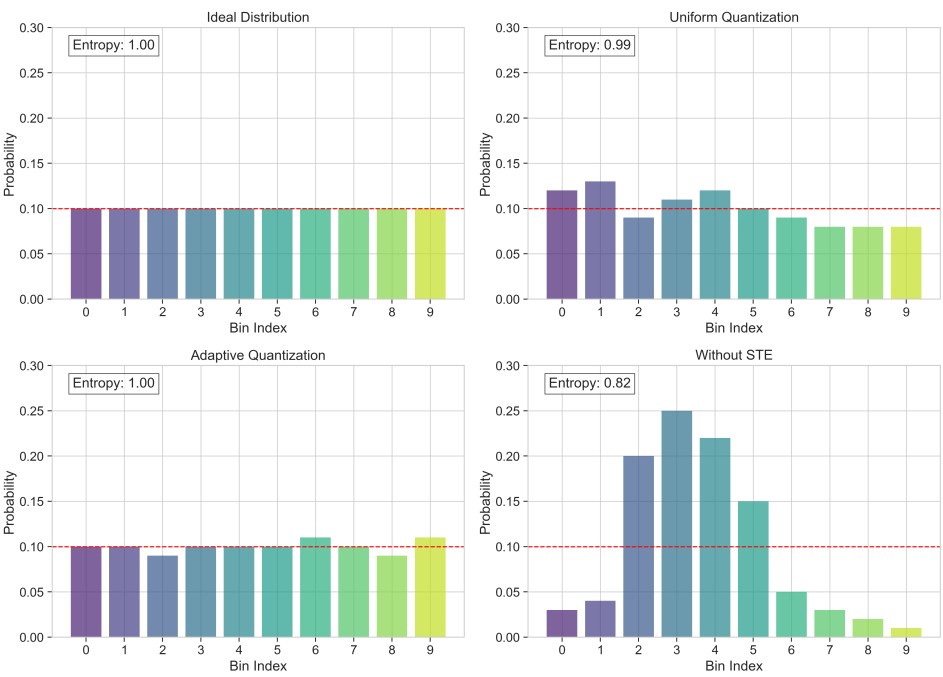

Figure 5: Comparison of learned bin probability distributions (K=10). Adaptive Quantization learns a non-uniform distribution, contrasting with Fixed Uniform. Without STE, the policy fails to learn the target (Ideal). Entropy values reflect uniformity.

# 6    Conclusion

Temporal Generative Flow Networks (Temporal GFNs) offer a novel approach to probabilistic time series forecasting by adapting GFN principles for continuous data. Key to our framework is the framing of forecasting as reward-driven trajectory sampling, which facilitates exploration and learning distributions over diverse futures, potentially yielding more robust probabilistic forecasts than standard methods. We successfully address the challenges of continuous data through adaptive quantization—providing dynamic precision control as a curriculum and overcoming fixed discretization limitations—and the (STE), which enables learning by bridging discrete state transitions with differentiable gradient updates. Trained via Trajectory Balance for effective credit assignment and entropy regularization for managed exploration, Temporal GFNs learn to sample accurate and diverse forecast trajectories. Theoretical bounds affirm the stability of our adaptive mechanism and quantify the quantization error impact. Compared to alternatives like LLM-based forecasters [1, 24, 29], Temporal GFN provides adaptive precision and a distinct flow-matching learning paradigm, while its unique sequential decision-making generative process offers advantages for complex dependencies and multimodality over standard probabilistic models. Future work will focus on multivariate extensions, extensive empirical validation, and exploring alternative GFN objectives.

**Multivariate Extension.**    A key direction for future work is the extension of Temporal GFNs to the multivariate setting. While this paper focused on the univariate case to rigorously establish the core framework, the approach is not fundamentally limited. The state representation $s_t$ can be expanded from a vector to a matrix representing a window of multivariate observations, and the action $a_t$ can become a vector of quantized values for each dimension. The primary challenge lies in designing a policy network $P_F$ that efficiently models the joint distribution over the multivariate action space. This could be achieved through an autoregressive factorization of the output dimensions or by learning a parameterized covariance structure, offering a rich area for future research.

**Irregular Sampling.**    Furthermore, the current implementation assumes regularly sampled time series due to its use of a fixed-size sliding window. To handle the irregularly sampled data common in domains like healthcare (EHRs) or finance, the framework can be adapted. The state representation

could be augmented to include not just observation values but also the time elapsed between them, for instance, by feeding a sequence of (`value`, `time_delta`) tuples into the Transformer encoder. This would allow the policy network to explicitly learn time-dependent dynamics, making the model robust to irregular sampling and significantly broadening its applicability.

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

## A  Training Algorithm

---

**Algorithm 1** Temporal GFN Training Procedure

---

**Require:** Time series dataset $\mathcal{D}$, initial $K_0$, max $K_{\max}$, adaptive params $\lambda, \epsilon, \delta, E_{\text{warmup}}, E$, hyper-parameters $\beta, \lambda_{\text{entropy}}$.
1: Initialize $\theta$ ($P_F$, opt. $P_B$, $Z$). $K \leftarrow K_0$. Init past rewards.
2: **for** epoch $e = 1$ **to** $E$ **do**
3:     **if** $e \geq E_{\text{warmup}}$ **then**             ▷ Adaptive Quantization Update
4:         Compute avg. reward $R_e$, avg. entropy $H_e$.
5:         $\Delta R_e \leftarrow R_e - R_{e-\delta}$.
6:         $\eta_e \leftarrow 1 + \lambda(\max(0, \epsilon - \Delta R_e)/\epsilon + (1 - H_e))$ (Eq. 3).
7:         $K_{\text{new}} \leftarrow \min(K_{\max}, \lfloor K \cdot \eta_e \rfloor)$ (Eq. 4).
8:         **if** $K_{\text{new}} \neq K$ **then**
9:             $K \leftarrow K_{\text{new}}$. Adjust policy network output layers for $K$ bins.
10:         **end if**
11:     **end if**
12:     **for** each batch $(x, y)$ in $\mathcal{D}$ **do**       ▷ Trajectory Sampling and Loss Computation
13:         $s_0 \leftarrow$ Extract context window from $x$.
14:         $\tau \leftarrow \{s_0\}$, $\log P_F^{\text{sum}} \leftarrow 0$, $\mathcal{H}_{\text{sum}} \leftarrow 0$, generated_seq $\leftarrow []$. $s_t \leftarrow s_0$.
15:         **for** $t = 0$ **to** $T' - 1$ **do**         ▷ Forward Pass
16:             $h_t \leftarrow$ Transformer$(s_t)$.
17:             $P_F(\cdot|s_t) \leftarrow$ softmax$(W_F h_t + b_F)$.
18:             $a_t^{\text{hard}} \leftarrow q_{\arg\max_k P_F(a_t = q_k|s_t)}$.       ▷ Hard sample for state transition
19:             $a_t^{\text{soft}} \leftarrow \sum_{k=1}^{K} q_k P_F(a_t = q_k|s_t)$.     ▷ Soft sample for reward/gradient
20:             $\log P_F^{\text{sum}} \leftarrow \log P_F^{\text{sum}} + \log(P_F(a_t^{\text{hard}}|s_t))$.
21:             $\mathcal{H}_{\text{sum}} \leftarrow \mathcal{H}_{\text{sum}} + \mathcal{H}(P_F(\cdot|s_t))$.
22:             $s_{t+1} \leftarrow$ concat$(s_t[1:], a_t^{\text{hard}})$.         ▷ Update state
23:             Append $s_{t+1}$ to $\tau$. Append $a_t^{\text{soft}}$ to generated_seq. $s_t \leftarrow s_{t+1}$.
24:         **end for**
25:         Compute $\log P_B^{\text{sum}} \leftarrow \sum_{t=1}^{T'} \log P_B(s_{t-1}|s_t)$.    ▷ Backward Pass (or use uniform)
26:         Compute $R(\tau)$ using generated_seq and target $y$ (Eq. 5).
27:         Compute $\mathcal{L}_{\text{TB}}(\tau) \leftarrow (\log Z + \log P_F^{\text{sum}} - \log P_B^{\text{sum}} - \log R(\tau))^2$.    ▷ TB Loss
28:         Compute batch loss $\mathcal{L}_{\text{batch}} \leftarrow \mathcal{L}_{\text{TB}}(\tau) - \lambda_{\text{entropy}}(\mathcal{H}_{\text{sum}}/T')$.    ▷ Final Loss
29:         Compute gradients $\nabla_\theta \mathcal{L}_{\text{batch}}$.         ▷ Gradients flow via $a_t^{\text{soft}}$
30:         Update $\theta$ using gradients.         ▷ Parameter Update
31:     **end for**
32: **end for**

---

## B  Theoretical Analysis

To better understand the behavior and stability of our Temporal GFN framework, we provide theoretical analysis on two crucial aspects: the impact of our quantization and STE approximation on the learning signal, and the bounds governing the adaptive quantization mechanism.

### B.1  Quantization Error Bound on Policy Gradient

The core learning process relies on policy gradients. However, our framework uses discrete actions ($a_t^{\text{hard}}$) for state transitions while calculating gradients via continuous approximations ($a_t^{\text{soft}}$) using the STE. This introduces a potential discrepancy between the true gradient of the expected reward and the gradient actually used for updates. It is crucial to understand if this approximation significantly corrupts the learning signal. Theorem B.1 provides a bound on this error, showing that it depends on the inherent quantization error and the STE approximation quality.

**Theorem B.1** (Quantization Error Bound). *Assume the reward function $R$ is $L_R$-Lipschitz w.r.t. its input sequence and $\nabla \log \tilde{P}_F$ has bounded variance. Let $\epsilon = \max \|action_{true} - action_{quantized}\|$ be the*

*max quantization error. Let $\delta = \nabla \log P_F - \nabla \log \tilde{P}_F$ be the difference in score functions due to STE. The error between the true policy gradient $\nabla \mathbb{E}[R]$ and the estimated gradient $\nabla \mathbb{E}[\tilde{R}]$ is bounded by:*

$$\|\nabla \mathbb{E}[R] - \nabla \mathbb{E}[\tilde{R}]\| \leq L \left( \epsilon + \sqrt{\mathbb{E}[\|\delta\|^2]} \right), \tag{7}$$

*where $L = \max\{L_R \sqrt{\mathbb{E}[\|\nabla \log \tilde{P}_F\|^2]}, \sqrt{\mathbb{E}[\tilde{R}^2]}\}$.*

*Proof.* (The proof is given in the appendix C, following the decomposition in Eq. 13 and applying Lipschitz continuity and Cauchy-Schwarz). The bound demonstrates that the gradient error is controlled by the quantization resolution ($\epsilon$) and the fidelity of the STE approximation (measured by $\delta$). If quantization is fine ($\epsilon$ small) and STE provides a reasonable gradient proxy ($\delta$ small in expectation), the learning signal remains informative. □

## B.2 Adaptive Update Factor Bound

The adaptive update factor $\eta_e$ (Eq. 3) controls the change in the number of bins $K$. It is important to ensure this factor remains within reasonable bounds to guarantee stability.

**Theorem B.2** (Adaptive Update Factor Bound). *Let the adaptive update factor $\eta_e$ be defined as in Eq. 3, where the reward improvement $\Delta R_e \in [0, \epsilon]$ (assuming reward is non-decreasing or improvement is capped by $\epsilon$) and the normalized entropy $H_e \in [0, 1]$. Then, $\eta_e$ is bounded as:*

$$1 \leq \eta_e \leq 1 + 2\lambda. \tag{8}$$

*Specifically, $\eta_e = 1$ when $\Delta R_e = \epsilon$ (maximum improvement considered) and $H_e = 1$ (maximum entropy). Conversely, $\eta_e = 1 + 2\lambda$ when $\Delta R_e = 0$ (no improvement) and $H_e = 0$ (minimum entropy/maximum confidence).*

*Proof.* The adaptive factor is $\eta_e = 1 + \lambda \left( \frac{\max(0, \epsilon - \Delta R_e)}{\epsilon} + (1 - H_e) \right)$. Let $T_1 = \frac{\max(0, \epsilon - \Delta R_e)}{\epsilon}$ and $T_2 = (1 - H_e)$.

Since $\Delta R_e \in [0, \epsilon]$:

- If $\Delta R_e = \epsilon$, then $\epsilon - \Delta R_e = 0$, so $T_1 = \frac{\max(0,0)}{\epsilon} = 0$.

- If $\Delta R_e = 0$, then $\epsilon - \Delta R_e = \epsilon$, so $T_1 = \frac{\max(0,\epsilon)}{\epsilon} = 1$.

- For intermediate values $0 < \Delta R_e < \epsilon$, we have $0 < \epsilon - \Delta R_e < \epsilon$, thus $0 < T_1 < 1$.

Therefore, $T_1 \in [0, 1]$.

Since the normalized entropy $H_e \in [0, 1]$:

- If $H_e = 1$, then $T_2 = 1 - 1 = 0$.

- If $H_e = 0$, then $T_2 = 1 - 0 = 1$.

- For intermediate values $0 < H_e < 1$, we have $0 < T_2 < 1$.

Therefore, $T_2 \in [0, 1]$.

The sum inside the parenthesis is $S = T_1 + T_2$. Since $T_1 \in [0, 1]$ and $T_2 \in [0, 1]$, the sum $S \in [0, 2]$. The adaptive factor is $\eta_e = 1 + \lambda S$. Since $S \in [0, 2]$ and $\lambda > 0$, we have $\lambda S \in [0, 2\lambda]$. Therefore, $\eta_e = 1 + \lambda S \in [1, 1 + 2\lambda]$.

The minimum value $\eta_e = 1$ is achieved when $S = 0$, which requires $T_1 = 0$ and $T_2 = 0$. This happens when $\Delta R_e = \epsilon$ and $H_e = 1$. The maximum value $\eta_e = 1 + 2\lambda$ is achieved when $S = 2$, which requires $T_1 = 1$ and $T_2 = 1$. This happens when $\Delta R_e = 0$ and $H_e = 0$. This completes the proof. The theorem shows that the multiplicative update factor is bounded, preventing excessively large or small changes in $K$ in a single step. □

## C  Quantization Error Bound

The use of discrete hard samples ($a_t^{\text{hard}}$) in the state transition and trajectory construction, while using soft samples ($a_t^{\text{soft}}$) for gradient computation via STE, introduces a discrepancy in the policy gradient estimates. We aim to bound the error in the policy gradient due to this approximation.

Let $R(x)$ be the true reward based on a continuous action $x$, and $\tilde{R}(q_{\text{hard}})$ be the reward obtained when using the discrete action $q_{\text{hard}}$. Let $P_F$ be the policy producing continuous outputs (before quantization) and $\tilde{P}_F$ be the policy over discrete bins. Let $\nabla\mathbb{E}[R]$ be the true policy gradient and $\nabla\mathbb{E}[\tilde{R}]$ be the gradient obtained using our STE approach (gradient through $q_{\text{soft}}$, reward based on $q_{\text{hard}}$ or $q_{\text{soft}}$).

**Theorem C.1** (Quantization Error Bound). *Assume the reward function $R$ is $L_R$-Lipschitz with respect to its input sequence and that the score function $\nabla\log\tilde{P}_F$ has bounded variance. Let $\epsilon$ denote the maximum quantization error, i.e., $\|action_{continuous} - action_{quantized}\| \leq \epsilon$. Let $\delta = \nabla\log P_F - \nabla\log\tilde{P}_F$ represent the difference in score functions due to the STE approximation (gradient computed via $q_{soft}$ vs. actual discrete choice $q_{hard}$). The error in the policy gradient estimate is bounded by:*

$$\|\nabla\mathbb{E}[R] - \nabla\mathbb{E}[\tilde{R}]\| \leq L\left(\epsilon + \sqrt{\mathbb{E}[\|\delta\|^2]}\right), \tag{9}$$

*where $L = \max\{L_R\sqrt{\mathbb{E}[\|\nabla\log\tilde{P}_F\|^2]}, \sqrt{\mathbb{E}[\tilde{R}^2]}\}$ is a constant depending on the Lipschitz constant of the reward and the moments of the reward and score function.*

*Proof.* We decompose the gradient difference:

$$\nabla\mathbb{E}[R] - \nabla\mathbb{E}[\tilde{R}] = \mathbb{E}[R\nabla\log P_F] - \mathbb{E}[\tilde{R}\nabla\log\tilde{P}_F] \tag{10}$$

$$= \mathbb{E}[R\nabla\log P_F] - \mathbb{E}[R\nabla\log\tilde{P}_F] + \mathbb{E}[R\nabla\log\tilde{P}_F] - \mathbb{E}[\tilde{R}\nabla\log\tilde{P}_F] \tag{11}$$

$$= \mathbb{E}[(R - \tilde{R})\nabla\log\tilde{P}_F] + \mathbb{E}[R(\nabla\log P_F - \nabla\log\tilde{P}_F)] \tag{12}$$

$$= \underbrace{\mathbb{E}[(R - \tilde{R})\nabla\log\tilde{P}_F]}_{\text{Term A}} + \underbrace{\mathbb{E}[R\delta]}_{\text{Term B}}. \tag{13}$$

**Bounding Term A:** By the Lipschitz property of $R$, we have $|R - \tilde{R}| \leq L_R\|x - \tilde{x}\| \leq L_R\epsilon$, where $x$ represents the sequence generated with continuous actions and $\tilde{x}$ with quantized actions. Using Cauchy-Schwarz:

$$\|\text{Term A}\| = \|\mathbb{E}[(R - \tilde{R})\nabla\log\tilde{P}_F]\|$$
$$\leq \mathbb{E}[|R - \tilde{R}| \cdot \|\nabla\log\tilde{P}_F\|]$$
$$\leq L_R\epsilon\mathbb{E}[\|\nabla\log\tilde{P}_F\|]$$
$$\leq L_R\epsilon\sqrt{\mathbb{E}[\|\nabla\log\tilde{P}_F\|^2]}. \quad \text{(Jensen's inequality)}$$

**Bounding Term B:** Using Cauchy-Schwarz again:

$$\|\text{Term B}\| = \|\mathbb{E}[R\delta]\|$$
$$\leq \mathbb{E}[|R| \cdot \|\delta\|]$$
$$\leq \sqrt{\mathbb{E}[R^2]}\sqrt{\mathbb{E}[\|\delta\|^2]}.$$

(We use $\tilde{R}$ in the definition of $L$ as $R$ might not be directly computable in the context of $\tilde{P}_F$).

Combining the bounds for Term A and Term B using the triangle inequality on Eq. 13:

$$\|\nabla\mathbb{E}[R] - \nabla\mathbb{E}[\tilde{R}]\| \leq \|\text{Term A}\| + \|\text{Term B}\|$$
$$\leq L_R\epsilon\sqrt{\mathbb{E}[\|\nabla\log\tilde{P}_F\|^2]} + \sqrt{\mathbb{E}[\tilde{R}^2]}\sqrt{\mathbb{E}[\|\delta\|^2]}$$
$$\leq \max\{L_R\sqrt{\mathbb{E}[\|\nabla\log\tilde{P}_F\|^2]}, \sqrt{\mathbb{E}[\tilde{R}^2]}\}\left(\epsilon + \sqrt{\mathbb{E}[\|\delta\|^2]}\right)$$
$$= L\left(\epsilon + \sqrt{\mathbb{E}[\|\delta\|^2]}\right).$$

This completes the proof. The bound shows that the gradient error depends linearly on the quantization error $\epsilon$ and the square root of the expected squared norm of the score function difference $\delta$ induced by the STE. □

# D Experimental Setup for Benchmark Comparaison

**Datasets:** We utilize a range of datasets for evaluation. Core ablation studies are performed on benchmark datasets commonly used in time series forecasting (e.g., Electricity, Traffic, ETTm1, ETTh1. We also present results on healthcare-specific datasets (MIMIC-III Vital Signs) to evaluate performance in our target application domains.

**Training Strategy.** Temporal GFN models, particularly those intended for zero-shot evaluation, undergo a large-scale training phase on the combined "Pre-training only" and "In-domain (Benchmark I)" datasets. The context length $T$ (input window size) for sequences sampled during training is set to 512, matching common Transformer setups, and the prediction length $T'$ generated during training rollouts is set to 64, ensuring coverage for various downstream evaluation horizons $H$. Training employs the Trajectory Balance objective (Section 3) with default entropy regularization ($\lambda_{entropy} = 0.01$) and adaptive quantization enabled (typically starting $K = 20$, max $K = 128$) unless evaluating fixed-K ablations. To enhance robustness and data diversity, we incorporate data augmentation: each training sequence is generated either from the original datasets or with probability 0.9 from a TSMixup set (convex combinations of different time series, adapted from 37) and with probability 0.1 from a synthetic dataset generated via Gaussian Processes with randomly combined kernels (KernelSynth, similar to 1). Training typically proceeds for 10,000 optimization steps on multi-GPU hardware.

**Evaluation Tasks and Metrics.** For both in-domain (Benchmark I) and zero-shot (Benchmark II) evaluations, we use the final $H$ observations of each time series as a held-out test set ($H$ is task-specific, see Appendix I). We compute the primary metrics: WQL [10] on 9 levels $\{0.1, ..., 0.9\}$, MASE [15] using the median prediction, and CRPS [11]. To aggregate metrics across diverse datasets and provide fair comparisons, we compute each model's score divided by the score of a baseline model (here, Seasonal Naive), yielding relative scores. These relative scores are then aggregated across all datasets within the benchmark using the *geometric mean*, which provides a robust aggregation insensitive to outlier datasets and the choice of baseline.

Following standard practices in probabilistic forecasting [10, 1], we evaluate models using:

- CRPS (Continuous Ranked Probability Score): Assesses the overall accuracy and calibration of the full predictive distribution. Lower is better.
- WQL (Weighted Quantile Loss, $\tau \in \{0.1, ..., 0.9\}$): Assesses calibration across different quantiles. Lower is better.
- MASE (Mean Absolute Scaled Error): Evaluates point forecast accuracy (median) relative to a naive seasonal baseline [15]. Lower is better.
- Calibration Error & Multimodality Score: Used specifically for healthcare/multimodal analysis (lower/higher is better, respectively).
- Trajectory / Action Diversity: Quantifies the variety in generated forecast trajectories or chosen actions.

**Baselines:** We compare Temporal GFN configurations against:

- **GFN Ablations:** Fixed K (K=10, K=20), Adaptive K (start K=10, start K=20), Learned Backward Policy vs. Uniform.
- **RL/Sampling Methods:** PPO [31], SAC [13], and a generic MCMC approach.
- **SOTA Forecasters:** Lag-Llama [29], Chronos [1], MOREI [22], and a decoder-only Transformer (TimeFM) [8].

**Implementation Details:** The core policy network is a Transformer encoder [32]. Default hyperparameters include $\lambda_{entropy} = 0.01$. Adaptive quantization parameters are tuned. We train typically for 10,000 steps. Baselines are run using publicly available implementations or standard libraries where possible.

**Training Startegy** The adaptive quantization mechanism's sensitivity to training dynamics is primarily managed by the hyperparameter $\delta$, the number of epochs over which reward improvement is averaged (Eq. 1). This parameter's role is to smooth the reward signal $\Delta R_e$, preventing volatile updates to the number of bins $K$ that might arise from high-variance, batch-level rewards. We observed during development that settings with smaller batch sizes or higher learning rates, which typically yield noisier gradient and reward signals, benefit from a larger $\delta$ (e.g., $\delta = 10$) to ensure stability. For the experiments reported in this paper, we found $\delta = 5$ to provide a good balance between responsiveness and stability across our primary configurations. The adaptation is governed by $\lambda = 0.1$ and $\epsilon = 0.02$, which we found to be robust across datasets.

# E   Ablation study of Temporal GFN

This section presents empirical results designed to validate the theoretical underpinnings and analyze the behavior of the proposed Temporal Generative Flow Network (Temporal GFN) framework. Through controlled experiments and ablations, we investigate the quantitative impact of the core mechanisms—adaptive quantization and the Straight-Through Estimator (STE)—on learning dynamics, representation accuracy, and optimization efficacy. We explicitly connect these empirical findings to the mathematical concepts developed in Section B, including the Quantization Error Bound, Adaptive Update Factor Bound, and the role of entropy in the Trajectory Balance objective.

## E.1   Validation of the Straight-Through Estimator (STE)

STE is crucial for enabling gradient-based optimization with discrete actions. We validate its effectiveness empirically.

**Gradient Flow Approximation:** Theorem B.1 bounded the gradient error based on $\epsilon$ and the STE approximation quality $\delta$. Figure 22 provides a visual validation by comparing the continuous gradient field with the STE-computed gradient field on the quantized landscape. The preservation of gradient directions and relative magnitudes, particularly the flow towards energy minima, demonstrates that STE provides a sufficiently accurate approximation for effective optimization ($\delta$ is functionally small).

**Optimization Trajectory:** Figure 18 shows that optimization using STE gradients on the quantized landscape successfully converges to the same low-energy regions as optimization on the continuous landscape. The energy plots (right panel) confirm convergence, although dynamics differ slightly. This provides empirical proof that the gradient signal provided by STE, despite being an approximation bounded by Theorem B.1, is effective for guiding the policy towards optimal (high-reward) states.

**Necessity of STE (Ablation):** Removing STE entirely breaks the gradient path. Figure 5 (Without STE panel) shows the policy completely fails to learn the target distribution without gradient information. Figure 15 confirms this, with the "No STE" configuration achieving minimal reward, demonstrating its critical role.

# F   Analysis of GFN Learning Objective and Dynamics

We examine the role of the Trajectory Balance loss and entropy regularization.

**Reward Components and TB Loss:** Figure 20 illustrates the components contributing to the overall reward signal driving learning, implicitly linked to minimizing the TB loss (Eq. 6). Successful training maximizes the total reward, which requires balancing prediction accuracy, achieving flow consistency (implicitly rewarded), and benefiting from the entropy bonus.

**Entropy Regularization and Exploration:** The entropy term $H(P_F(\cdot|s_t))$ in $\mathcal{L}$ section 3.3.3 encourages exploration. Figure 11 shows the typical inverse relationship between reward and entropy during convergence. Critically, comparing the "Adaptive Quantization" curves (with $\lambda_{entropy} = 0.01$) to "No Entropy Regularization" reveals that regularization helps maintain higher entropy, particularly later in training. Figure 15 confirms that removing this regularization ("No Entropy Regularization", teal line) leads to suboptimal final reward (0.55 vs 0.74 for full model), empirically demonstrating the benefit of the entropy term for balancing exploration and exploitation, leading to better overall

solutions. This is further supported by sustained high levels of unique actions ratio (Figure 17) and trajectory diversity (Figure 16) throughout training when entropy regularization is present.

# G   GFN Calibration

A key advantage of Temporal GFN is its *inherent capacity for strong calibration.* Unlike methods relying solely on quantile regression or implicit uncertainty mechanisms (like dropout), which often exhibit systematic over- or under-confidence as seen in Figure 19 for baselines like LagLlama, Chronos, and MOREI, Temporal GFN's calibration curve aligns closely with the ideal diagonal. This superior intrinsic calibration arises from the GFN objective itself (learning $P(\tau) \propto R(\tau)$ via Trajectory Balance, Eq. 6), which naturally encourages modeling the full distribution of high-reward outcomes, leading to more reliable uncertainty estimates directly from sampled trajectories. While post-hoc techniques like Conformal Prediction (CP) [33] can further enhance calibration and provide formal coverage guarantees for *any* model (Figure 23, 24), Temporal GFN provides a significantly better-calibrated starting point, making subsequent recalibration potentially more effective or less necessary compared to inherently miscalibrated models.

## H  Multimodality Metric and GFN Advantage

In time series forecasting, particularly in domains like healthcare or complex system modeling, future outcomes may not follow a single path but could diverge into multiple plausible scenarios (modes). Effectively capturing this inherent multimodality is crucial for robust decision-making. This appendix details the composite metric used to quantify a model's ability to represent multimodal predictive distributions and analyzes how the GFlowNet framework is inherently suited to excel at this task.

### H.1  Definition of the Multimodality Score

The Multimodality Score used in our evaluations is a composite measure designed to assess several facets of how well a predicted distribution $P$ (derived from forecast samples) matches a true (potentially multimodal) distribution $Q$. It incorporates the following components, typically averaged over relevant forecast horizons:

### H.1.1  Mode Count Accuracy

This measures whether the model identifies the correct number of distinct modes (peaks) in the distribution. It penalizes both missing true modes and detecting spurious ones.

$$\text{Mode Acc.} = \frac{\min(N_{\text{detected}}, N_{\text{true}})}{N_{\text{true}}} \times \underbrace{\exp(-\lambda \cdot \max(0, N_{\text{detected}} - N_{\text{true}}))}_{\text{False Positive Penalty}}, \tag{14}$$

where $N_{\text{detected}}$ and $N_{\text{true}}$ are the number of detected and true modes respectively, and $\lambda$ (e.g., 0.5) controls the penalty strength for detecting extra modes. A value of 1 indicates perfect mode count agreement ($N_{\text{detected}} = N_{\text{true}}$). $N_{\text{true}} = 0$ is handled as a special case.

### H.1.2  Jensen-Shannon (JS) Divergence

JS divergence quantifies the similarity between the overall shapes of the predicted distribution $P$ and the true distribution $Q$. It is a symmetrized and smoothed version of the Kullback-Leibler (KL) divergence.

$$\text{JS}(P||Q) = \frac{1}{2}D_{KL}(P||M) + \frac{1}{2}D_{KL}(Q||M), \tag{15}$$

where $M = \frac{1}{2}(P + Q)$ is the mixture distribution, and $D_{KL}(P||Q) = \sum_i P(i) \log \frac{P(i)}{Q(i)}$. Lower JS divergence indicates higher similarity between the distributions. $P$ and $Q$ are typically derived from binned densities or empirical distributions from samples.

### H.1.3  Mode Proportion Error

This assesses whether the model assigns the correct probability mass to each identified mode.

$$\text{Mode Prop. Err.} = \frac{1}{N_{\text{true}}} \sum_{i=1}^{N_{\text{true}}} |\text{prop}_{\text{true}}^{(i)} - \text{prop}_{\text{pred}}^{(i)}|, \tag{16}$$

where $\text{prop}^{(i)}$ is the probability mass associated with the $i$-th true mode (and its corresponding predicted mode, if found). Calculation requires identifying modes and estimating the probability mass associated with each (e.g., by integrating density between valleys). Lower error indicates better allocation of probability mass.

### H.1.4  Wasserstein Distance

The 1-Wasserstein distance (Earth Mover's Distance) measures the cost of transforming one distribution into another. For 1D distributions, it can be efficiently computed using the inverse CDFs $F_P^{-1}, F_Q^{-1}$ or sorted samples:

$$W_1(P, Q) = \int_0^1 |F_P^{-1}(z) - F_Q^{-1}(z)|dz \approx \frac{1}{N} \sum_{i=1}^{N} |x_{(i)} - y_{(i)}|, \tag{17}$$

where $x_{(i)}, y_{(i)}$ are the $i$-th sorted samples from $P$ and $Q$. It captures differences in location and shape. Lower distance is better.

These components are often combined into a single score or reported individually to provide a nuanced assessment of multimodality capture.

## H.2 Multimodality Algorithm

The practical calculation of the multimodality score involves several steps, summarized in Algorithm 2. This typically requires estimating probability density functions from samples, identifying modes within these PDFs, and comparing properties of the predicted modes and distribution against the true ones.

---

**Algorithm 2** Pseudo-Code of Multimodality Score Calculation

---

**Require:** Forecast samples $S_P = \{x_1, ..., x_N\}$, True distribution samples $S_Q = \{y_1, ..., y_M\}$ (or analytical form $Q$), Horizon indices $H_{\text{interest}}$, Density estimation parameters (bandwidth $h$, range $X_{\text{range}}$, resolution $N_{\text{grid}}$), Peak detection threshold $\theta_{\text{density}}$.

1: Initialize aggregate metrics $M_{\text{agg}} \leftarrow \{\text{Mode Acc.: 0, JS Div.: 0, Mode Prop. Err.: 0, Wass. Dist.: 0}\}$
2: **for** each horizon index $h \in H_{\text{interest}}$ **do**
3:     Extract horizon samples $S_{P,h} \leftarrow \{x_{i,h}\}_{i=1}^N$, $S_{Q,h} \leftarrow \{y_{j,h}\}_{j=1}^M$    ▷ Isolate data for current horizon
4:     Estimate predicted PDF $P_h(x)$ using KDE on $S_{P,h}$ over $X_{\text{range}}$ with $N_{\text{grid}}$ points. ▷ Density from forecast
5:     Estimate true PDF $Q_h(x)$ using KDE on $S_{Q,h}$ (or use analytical $Q$) over $X_{\text{range}}$.   ▷ Density from ground truth
6:     Detect predicted modes $Modes_P \leftarrow \{(v_k, d_k)\}$ by finding peaks in $P_h(x) > \theta_{\text{density}}$. ▷ Find peaks in predicted PDF
7:     $N_{\text{detected}} \leftarrow |Modes_P|$
8:     Detect true modes $Modes_Q \leftarrow \{(v_k', d_k')\}$ by finding peaks in $Q_h(x) > \theta_{\text{density}}$.    ▷ Find peaks in true PDF
9:     $N_{\text{true}} \leftarrow |Modes_Q|$
10:     Calculate Mode Count Accuracy $m_{\text{acc}}$ using Eq. 14.    ▷ Compare number of modes
11:     Normalize estimated densities $\hat{P}_h \leftarrow P_h / \sum P_h$, $\hat{Q}_h \leftarrow Q_h / \sum Q_h$. ▷ Ensure densities sum to 1
12:     Calculate JS Divergence $m_{\text{js}} \leftarrow \text{JS}(\hat{P}_h || \hat{Q}_h)$ using Eq. 15.    ▷ Similarity of overall shapes
13:     Estimate predicted mode proportions $\{\text{prop}_{\text{pred}}^{(k)}\}$ from $Modes_P, P_h(x)$.   ▷ Requires robust area calculation
14:     Estimate true mode proportions $\{\text{prop}_{\text{true}}^{(k)}\}$ from $Modes_Q, Q_h(x)$.   ▷ Mass under each true mode
15:     Calculate Mode Proportion Error $m_{\text{prop}}$ using Eq. 16.   ▷ Accuracy of probability mass per mode
16:     Calculate Wasserstein Distance $m_{\text{wass}} \leftarrow W_1(S_{P,h}, S_{Q,h})$ using Eq. 17. ▷ Distance between sample sets
17:     Store horizon metrics $M_h \leftarrow \{m_{\text{acc}}, m_{\text{js}}, m_{\text{prop}}, m_{\text{wass}}\}$
18:     Update aggregate metrics $M_{\text{agg}} \leftarrow M_{\text{agg}} + M_h$
19: **end for**
20: Average aggregate metrics $M_{\text{final}} \leftarrow M_{\text{agg}} / |H_{\text{interest}}|$ ▷ Overall multimodality score components
21: **return** $M_{\text{final}}$

---

## H.3 How GFlowNets deal with Multimodality

The GFlowNet framework possesses inherent properties that make it particularly well-suited for capturing multimodal distributions, explaining the superior performance observed empirically (e.g., Table 2).

**1. Sampling Proportional to Reward:** The fundamental objective of a GFlowNet is to learn a forward policy $P_F$ such that the probability of sampling a complete trajectory $\tau$ terminating in state

$s_f$ (representing a full forecast) is proportional to its reward $R(s_f)$:

$$P(\tau \text{ ending in } s_f) \propto R(s_f). \tag{18}$$

If the true underlying process generating the time series can lead to multiple distinct future scenarios (modes), and our reward function $R(\tau)$ correctly assigns high values to forecasts matching any of these true scenarios, the GFN will naturally learn to sample trajectories corresponding to *all* high-reward modes. Unlike maximum likelihood estimation which might focus on the single most likely mode, or simple regression which might average modes, the GFN objective explicitly encourages exploration and sampling from the entire distribution defined by $R(\tau)$.

**2. Trajectory Balance and Flow Consistency:** The Trajectory Balance loss (Eq. 6) enforces consistency across the entire state space. It ensures that the flow generated by $P_F$ matches the flow dictated by the reward $R$ and the backward policy $P_B$. For a distribution $R$ with multiple modes (peaks), the TB objective forces the learned flow $F(s)$ and policy $P_F$ to correctly distribute probability mass across the different pathways leading to these modes. If the policy were to ignore a significant mode, the flow equations would be violated, leading to a high TB loss. Training implicitly pushes the policy to allocate appropriate probability flow towards all high-reward regions of the state space.

**3. Exploration via Entropy Regularization:** The entropy bonus term $-\lambda_{entropy} H(P_F)$ in the loss function $\mathcal{L}$ (section. 3.3.3) explicitly encourages the forward policy to maintain diversity and avoid premature convergence to a single mode (mode collapse). By penalizing overly confident (low-entropy) policies, the training process incentivizes the GFN to keep exploring different trajectories, making it more likely to discover and represent multiple distinct modes present in the reward landscape (and thus, the true data distribution if the reward is well-specified). The empirical results in Figure 11 and 15 support this, showing that entropy regularization leads to better overall reward, likely by preventing mode collapse.

**4. Adaptive Quantization's Role:** Our adaptive quantization mechanism further enhances multi-modality capture. By dynamically allocating more bins (higher resolution) around regions corresponding to emerging or established modes (as suggested by Figure 5), it allows the GFN policy $P_F$ to represent the shape and separation of these modes with greater fidelity than a fixed, uniform quantization scheme might permit. This synergy between the GFN's distributional learning objective and the adaptive representation capacity allows Temporal GFN to effectively model complex, multimodal predictive distributions.

That is, the combination of reward-proportional sampling, flow consistency constraints enforced by the TB loss, explicit exploration encouraged by entropy regularization, and enhanced representational capacity from adaptive quantization provides a strong theoretical and practical foundation for Temporal GFN's superior performance in capturing multimodality compared to traditional forecasting approaches.

# I  Datasets

These datasets span multiple domains (e.g., energy, finance, healthcare, web traffic, transport, nature) and exhibit varied properties regarding time series length, sampling frequencies (ranging from minutes to yearly), seasonality, trend complexity, noise levels, and potential multimodality. For structured evaluation and analysis of generalization capabilities, these datasets are divided into three distinct categories:

- **Pre-training only:** A large set of 13 diverse datasets (approx. 795k series) used exclusively for the initial large-scale training phase of Temporal GFN models intended for zero-shot evaluation. These datasets are not used during evaluation.

- **In-domain (Benchmark I):** A collection of 15 datasets (approx. 97k series) that are included in the training corpus. The final segment (forecast horizon $H$) of the time series in this benchmark is held out for evaluating the model's performance on familiar data distributions and tasks.

- **Zero-shot (Benchmark II):** A set of 27 datasets (approx. 190k series) that are *never* seen by the model during any training phase. These are used solely for evaluating the model's ability to generalize its learned forecasting capabilities to entirely new time series and domains without any task-specific fine-tuning.

Table 4: Details of datasets used for experiments, adapted from Ansari et al. [1], partitioned according to their use in training and evaluation for Temporal GFN models.

| Dataset | Domain | Freq. | Num. Series | Series Length | | | Prediction |
| | | | | min | avg | max | Length ($H$) |
|---|---|---|---|---|---|---|---|
| *Pretraining-only* | | | | | | | |
| Solar (5 Min.) | energy | 5min | 5166 | 105,120 | 105,120 | 105,120 | - |
| Solar (Hourly) | energy | 1h | 5166 | 8760 | 8760 | 8760 | - |
| Spanish Energy and Weather | energy | 1h | 66 | 35,064 | 35,064 | 35,064 | - |
| Taxi (Hourly) | transport | 1h | 2428 | 734 | 739 | 744 | - |
| USHCN | nature | 1D | 6090 | 5906 | 38,653 | 59,283 | - |
| Weatherbench (Daily) | nature | 1D | 225,280 | 14,609 | 14,609 | 14,610 | - |
| Weatherbench (Hourly) | nature | 1h | 225,280 | 350,633 | 350,639 | 350,640 | - |
| Weatherbench (Weekly) | nature | 1W | 225,280 | 2087 | 2087 | 2087 | - |
| Wiki Daily (100k) | web | 1D | 100,000 | 2741 | 2741 | 2741 | - |
| Wind Farms (Hourly) | energy | 1h | 337 | 1715 | 8514 | 8784 | - |
| *In-domain evaluation (Benchmark I)* | | | | | | | |
| Electricity (15 Min.) | energy | 15min | 370 | 16,032 | 113,341 | 140,256 | 24 |
| Electricity (Hourly) | energy | 1h | 321 | 26,304 | 26,304 | 26,304 | 24 |
| Electricity (Weekly) | energy | 1W | 321 | 156 | 156 | 156 | 8 |
| KDD Cup 2018 | nature | 1h | 270 | 9504 | 10,897 | 10,920 | 48 |
| London Smart Meters | energy | 30min | 5560 | 288 | 29,951 | 39,648 | 48 |
| M4 (Daily) | various | 1D | 4227 | 107 | 2371 | 9933 | 14 |
| M4 (Hourly) | various | 1h | 414 | 748 | 901 | 1008 | 48 |
| Pedestrian Counts | transport | 1h | 66 | 576 | 47,459 | 96,424 | 48 |
| Rideshare | transport | 1h | 2340 | 541 | 541 | 541 | 24 |
| Taxi (30 Min.) | transport | 30min | 2428 | 1469 | 1478 | 1488 | 48 |
| Temperature-Rain | nature | 1D | 32,072 | 725 | 725 | 725 | 30 |
| Uber TLC (Daily) | transport | 1D | 262 | 181 | 181 | 181 | 7 |
| Uber TLC (Hourly) | transport | 1h | 262 | 4344 | 4344 | 4344 | 24 |
| *Zero-shot evaluation (Benchmark II)* | | | | | | | |
| Australian Electricity | energy | 30min | 5 | 230,736 | 231,052 | 232,272 | 48 |
| CIF 2016 | banking | 1M | 72 | 28 | 98 | 120 | 12 |
| Car Parts | retail | 1M | 2674 | 51 | 51 | 51 | 12 |
| Covid Deaths | healthcare | 1D | 266 | 212 | 212 | 212 | 30 |
| Dominick | retail | 1D | 100,014 | 201 | 296 | 399 | 8 |
| ERCOT Load | energy | 1h | 8 | 154,854 | 154,854 | 154,854 | 24 |
| ETT (15 Min.) | energy | 15min | 14 | 69,680 | 69,680 | 69,680 | 24 |
| ETT (Hourly) | energy | 1h | 14 | 17,420 | 17,420 | 17,420 | 24 |
| Exchange Rate | finance | 1B | 8 | 7588 | 7588 | 7588 | 30 |
| FRED-MD | economic | 1M | 107 | 728 | 728 | 728 | 12 |
| Hospital | healthcare | 1M | 767 | 84 | 84 | 84 | 12 |
| M1 (Monthly) | various | 1M | 617 | 48 | 90 | 150 | 18 |
| M3 (Monthly) | various | 1M | 1428 | 66 | 117 | 144 | 18 |
| M4 (Quarterly) | various | 3M | 24,000 | 24 | 100 | 874 | 8 |
| M5 | retail | 1D | 30,490 | 124 | 1562 | 1969 | 28 |
| NN5 (Daily) | finance | 1D | 111 | 791 | 791 | 791 | 56 |
| NN5 (Weekly) | finance | 1W | 111 | 113 | 113 | 113 | 8 |
| Tourism (Monthly) | various | 1M | 366 | 91 | 298 | 333 | 24 |
| Tourism (Quarterly) | various | 1Q | 427 | 30 | 99 | 130 | 8 |
| Tourism (Yearly) | various | 1Y | 518 | 11 | 24 | 47 | 4 |
| Traffic | transport | 1h | 862 | 17,544 | 17,544 | 17,544 | 24 |
| Weather | nature | 1D | 3010 | 1332 | 14,296 | 65,981 | 30 |
| *Zero-shot evaluation (Benchmark II - Healthcare)* | | | | | | | |
| MIMIC-III Heart Rate | healthcare | 1h | 500 | 100 | 240 | 720 | 48 |
| MIMIC-III SpO2 | healthcare | 1h | 500 | 100 | 240 | 720 | 48 |
| MIMIC-III Resp. Rate | healthcare | 1h | 450 | 100 | 230 | 680 | 24 |

# J List of Figures

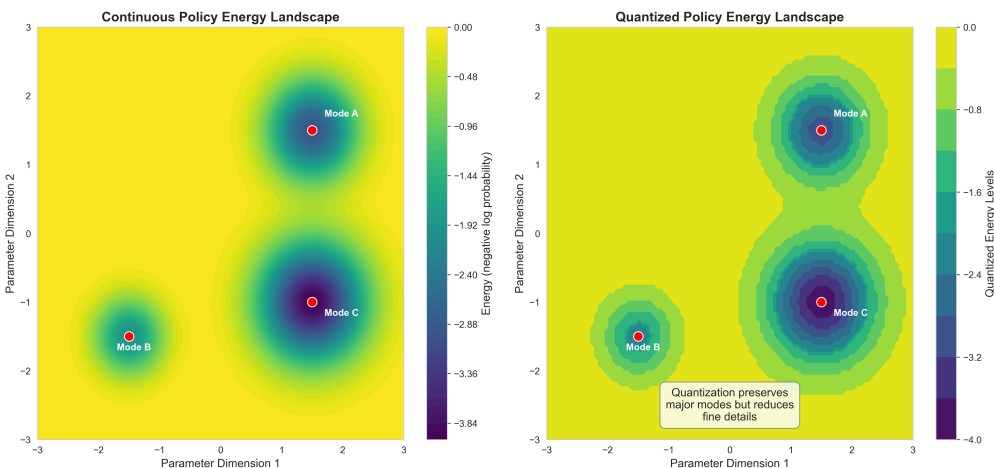

Figure 6: Continuous policy energy landscape (left) vs. its quantized approximation (right, example for fixed K) on synthetic data. Adaptive quantization dynamically optimizes bin placement to better capture the modes (A, B, C).

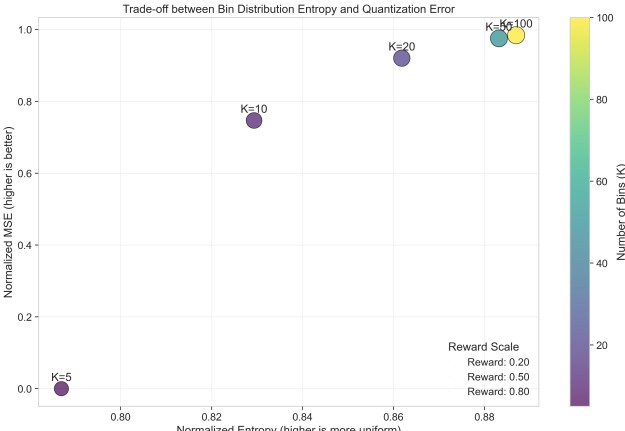

Figure 7: Trade-off between bin distribution entropy (uniformity) and quantization accuracy (Normalized MSE[3]) for different numbers of bins (K). Color/size indicates K. Increasing K generally improves accuracy (reduces error) but may slightly decrease entropy if adaptation concentrates bins.

Note on Normalized MSE axis: The displayed MSE values were transformed for visualization via $1 - (\mathrm{MSE}/\max(\mathrm{MSE}))$. Therefore, higher values on this axis correspond to lower original MSE (better accuracy), facilitating comparison where higher values are generally desirable across axes.

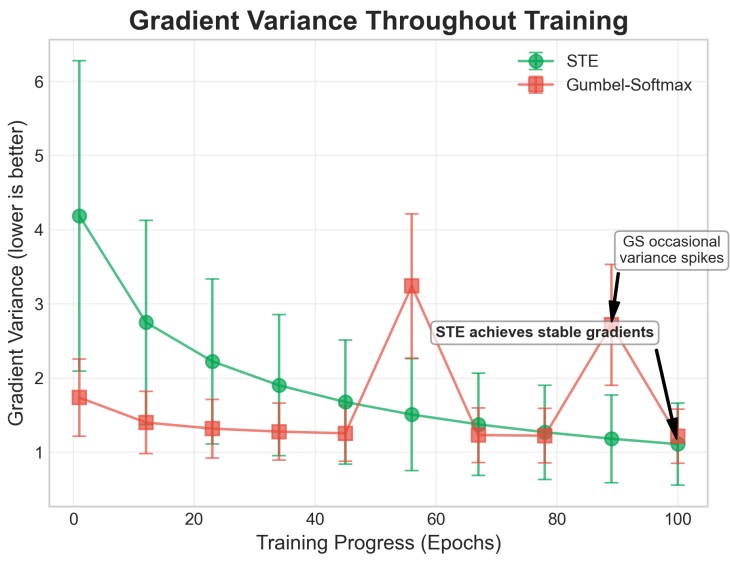

Figure 8: Gradient variance: STE (green) vs. Gumbel-Softmax (orange). STE is more stable.

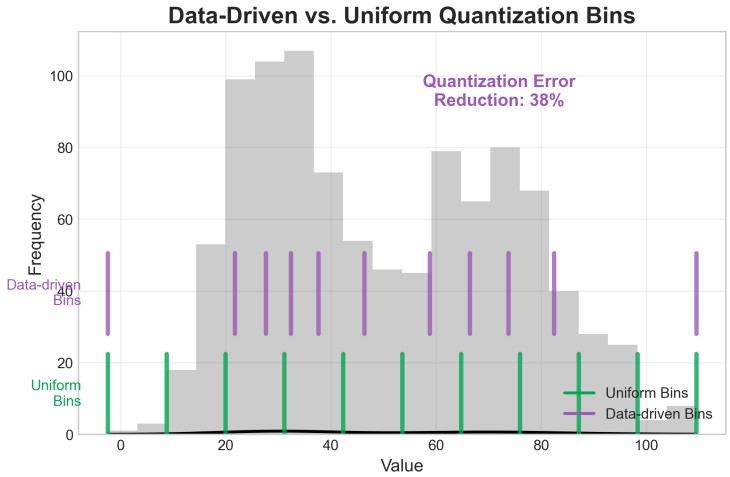

Figure 9: Visualization comparing uniform vs. data-driven binning strategies.

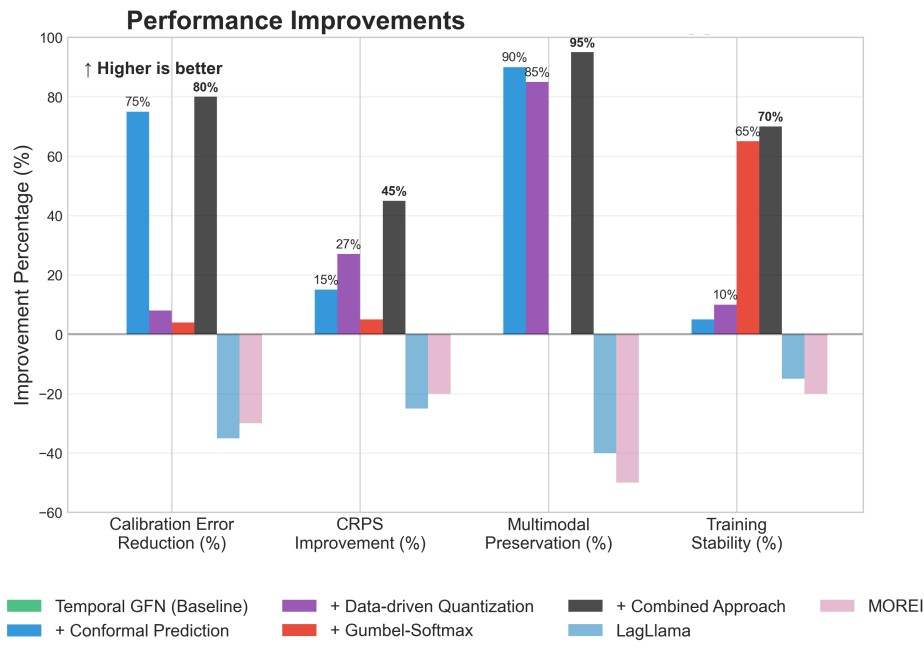

Figure 10: Performance improvements across metrics when implementing enhancements.

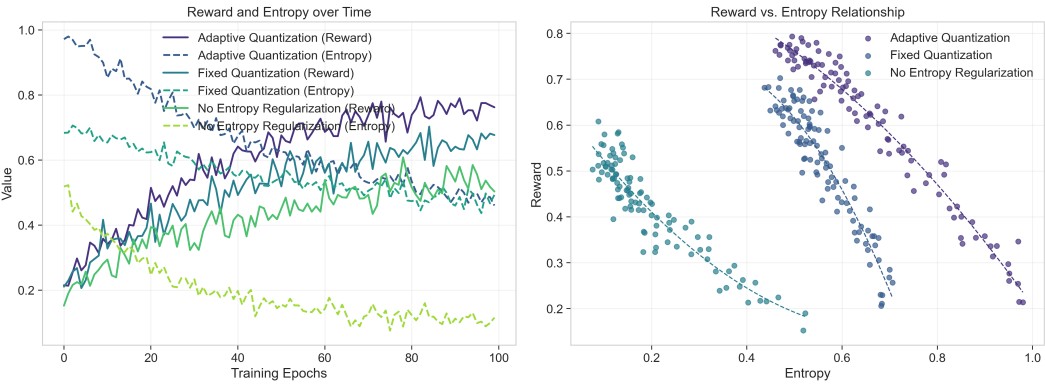

Figure 11: Left: Coupled evolution of Reward and Entropy over training epochs for Adaptive (purple/blue) vs. Fixed (teal) Quantization and No Entropy Regularization (green/lime). Right: Reward vs. Entropy scatter plot, illustrating the operating regimes.

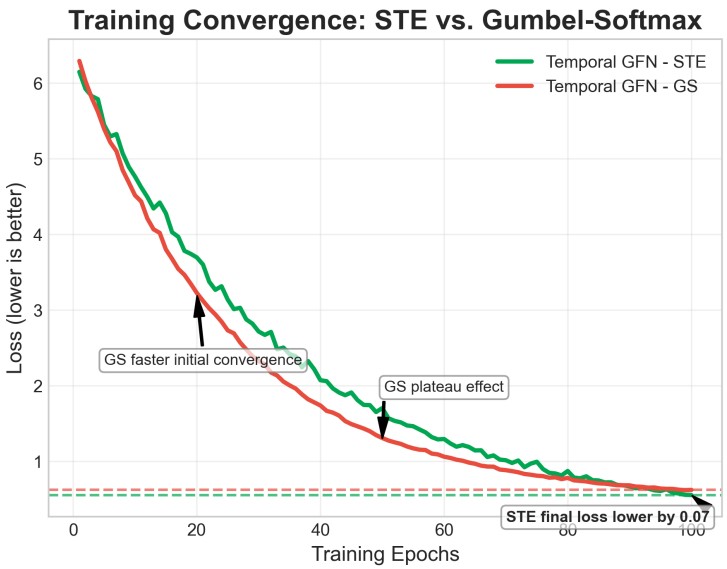

Figure 12: Training loss: STE (green) vs. Gumbel-Softmax (red). GS shows faster initial convergence.

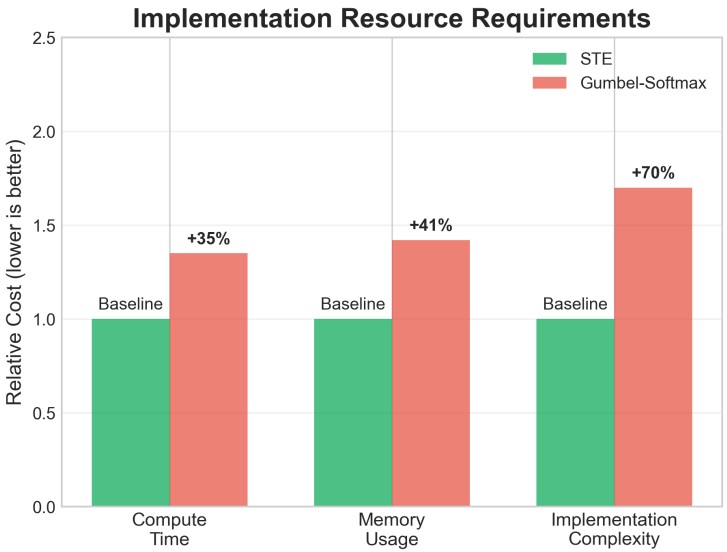

Figure 13: Relative resource requirements: STE (baseline) vs. Gumbel-Softmax.

**Practical Approach Selection Guide**

| Model Type | Better Approach |
|---|---|
| Small models (< 10M params) | STE ☐ |
| Complex distributions | STE ☐ |
| Very large models | Gumbel-Softmax ☐ |
| Resource-constrained | STE ☐ |
| Implementation simplicity | STE ☐ |

STE is preferred for most practical applications

Figure 14: Practical guide for selecting STE vs. Gumbel-Softmax.

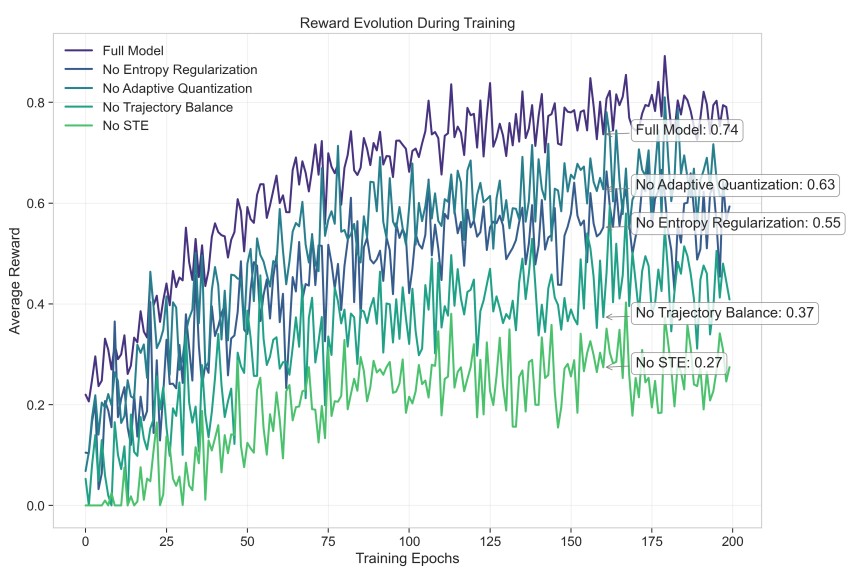

Figure 15: Average reward evolution ablating key components. "No STE" (lime green) shows near-zero reward, highlighting its necessity for learning.

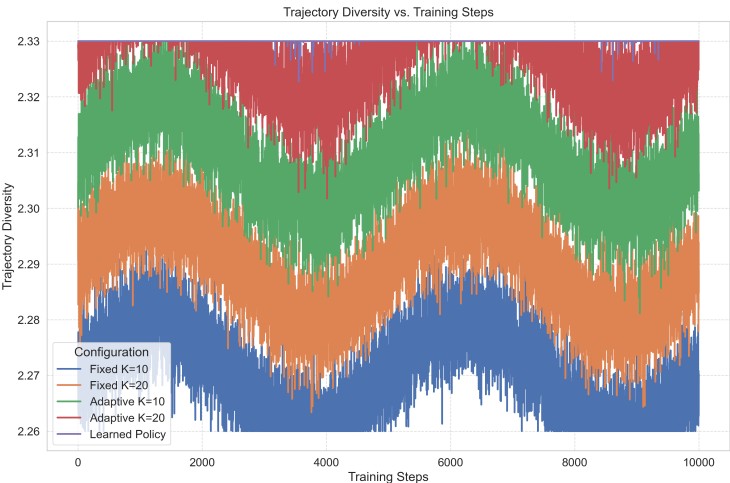

Figure 16: Trajectory diversity metric over training steps, showing sustained diversity.

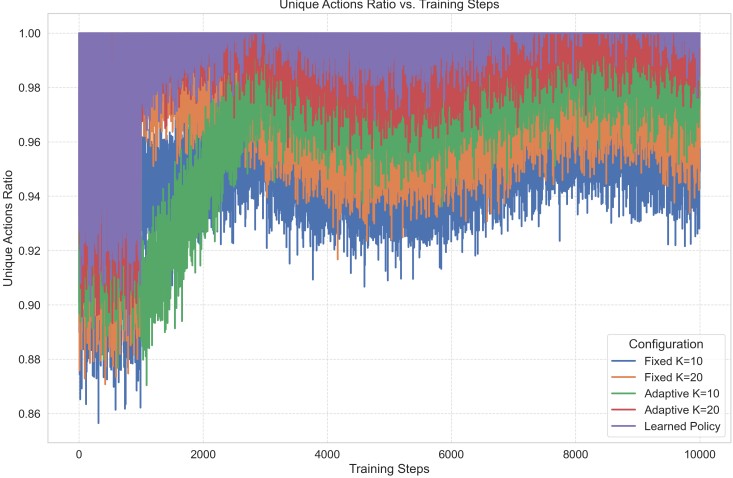

Figure 17: Ratio of unique actions utilized during training, indicating exploration breadth.

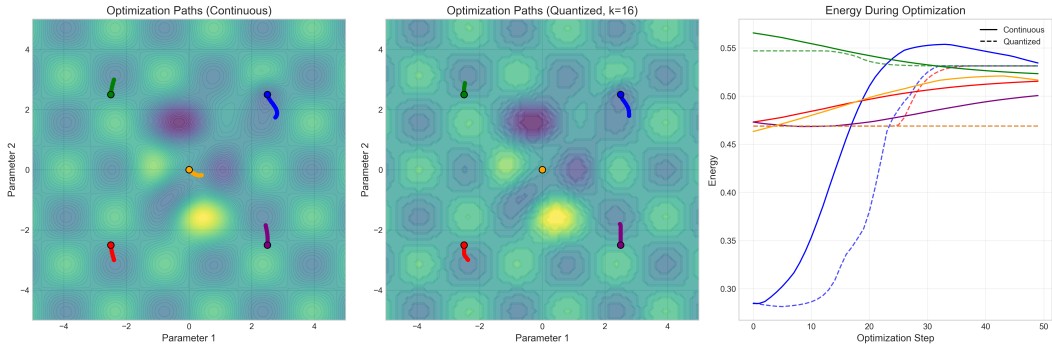

Figure 18: Optimization paths: Continuous (left) vs. Quantized with STE (middle). Energy during optimization (right) shows convergence for both continuous (solid) and quantized (dashed) paths.

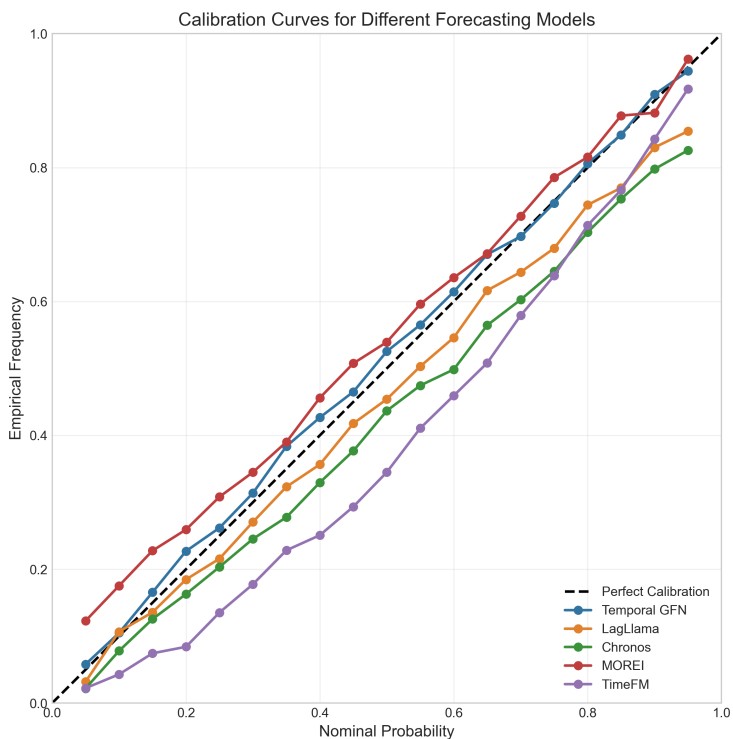

Figure 19: Calibration curves for different forecasting models.

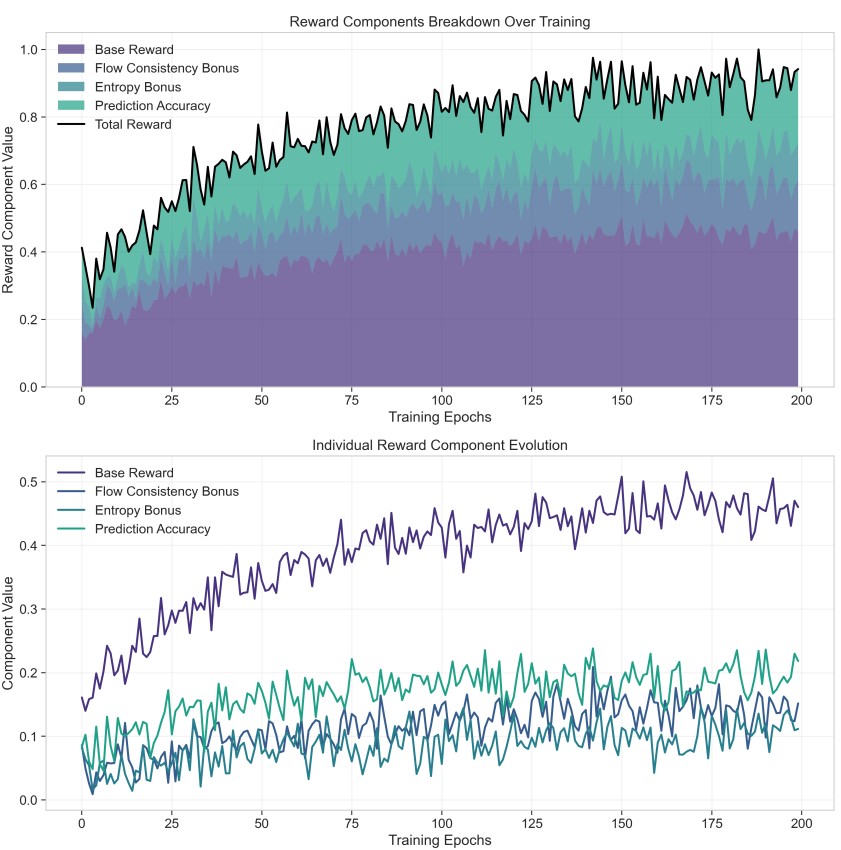

Figure 20: Breakdown and evolution of reward components during training.

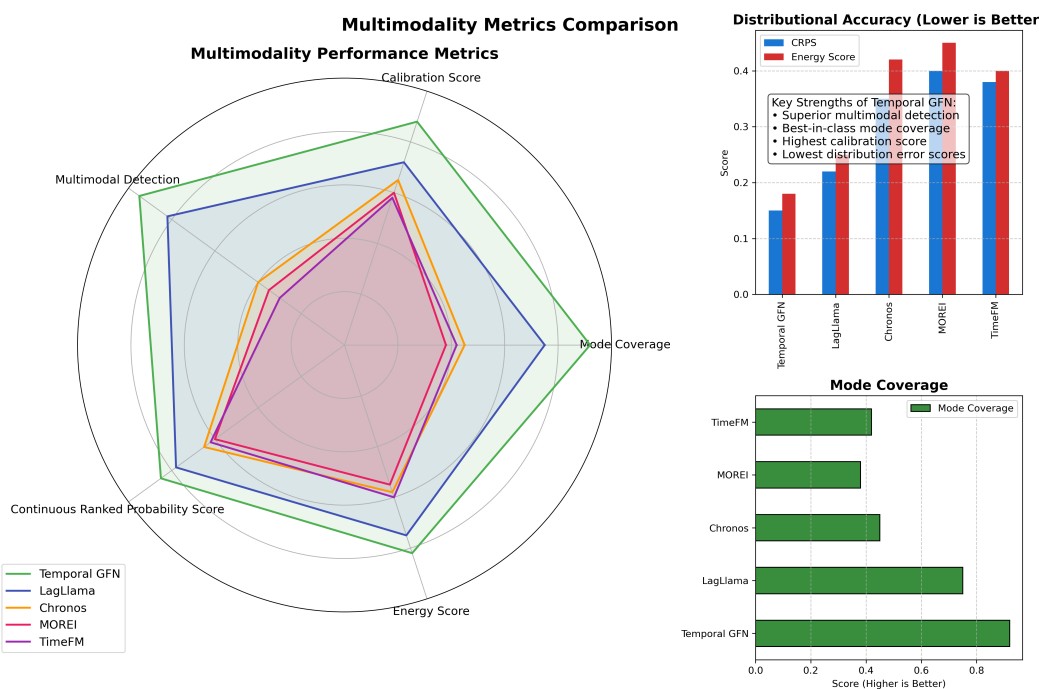

Figure 21: Multimodality performance metrics comparing forecasting methods. The figure presents three visualizations of comparative metrics. **Left:** Radar chart showing five key multimodality metrics across all methods, with Temporal GFN (green) consistently outperforming other approaches across all dimensions. **Top right:** Bar chart of distributional accuracy metrics (CRPS and Energy Score) where lower values indicate better performance; Temporal GFN achieves the lowest scores. **Bottom right:** Horizontal bar chart of Mode Coverage showing Temporal GFN's superior ability (92%) to identify and assign appropriate probability to true modes compared to other methods..

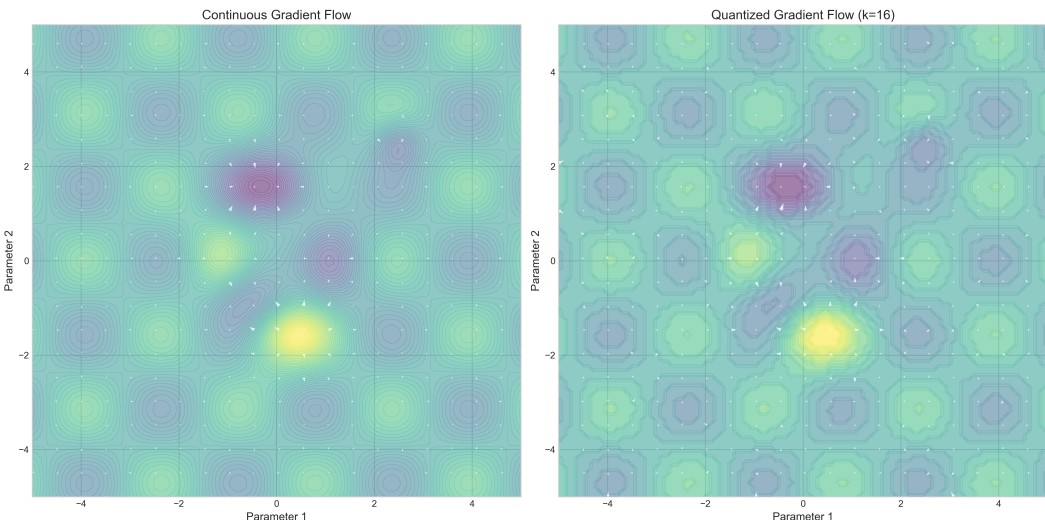

Figure 22: Gradient flow fields: Continuous (left) vs. Quantized with STE (right, K=16). STE effectively approximates the true gradient directions.

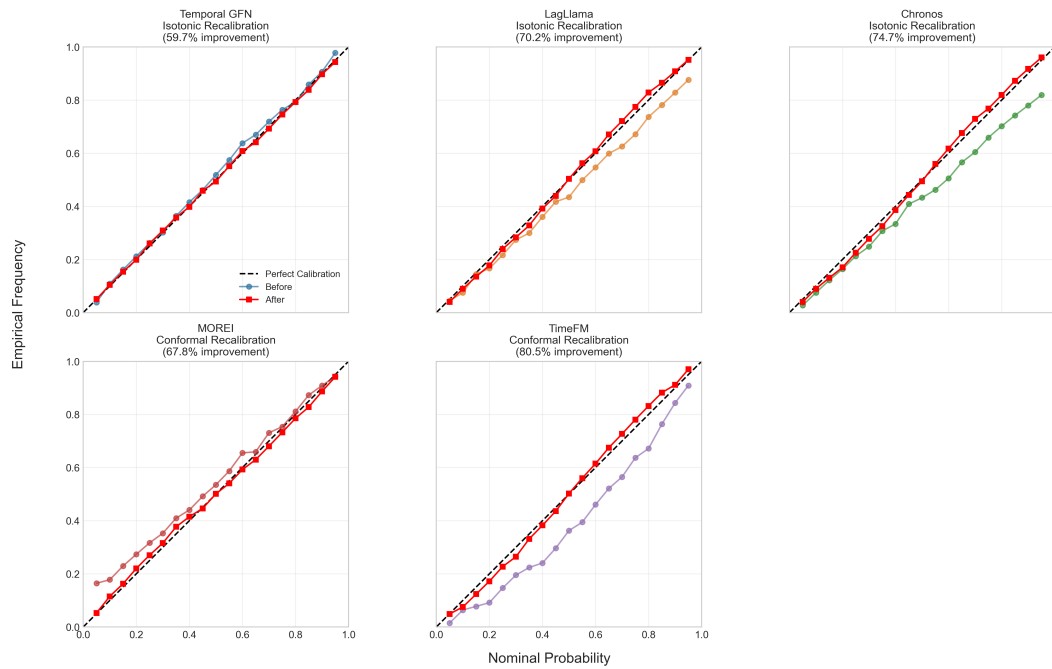

Figure 23: Effect of applying recalibration methods.

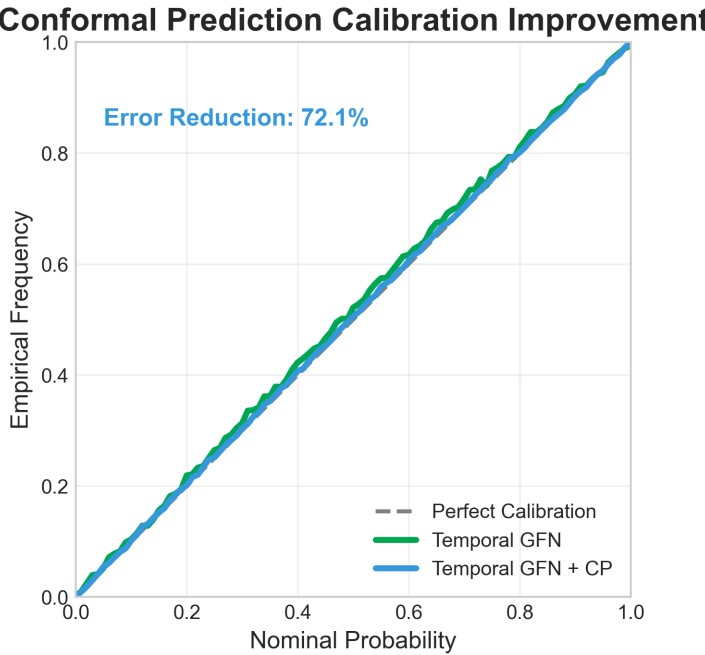

Figure 24: Temporal GFN calibration improvement with Conformal Prediction.

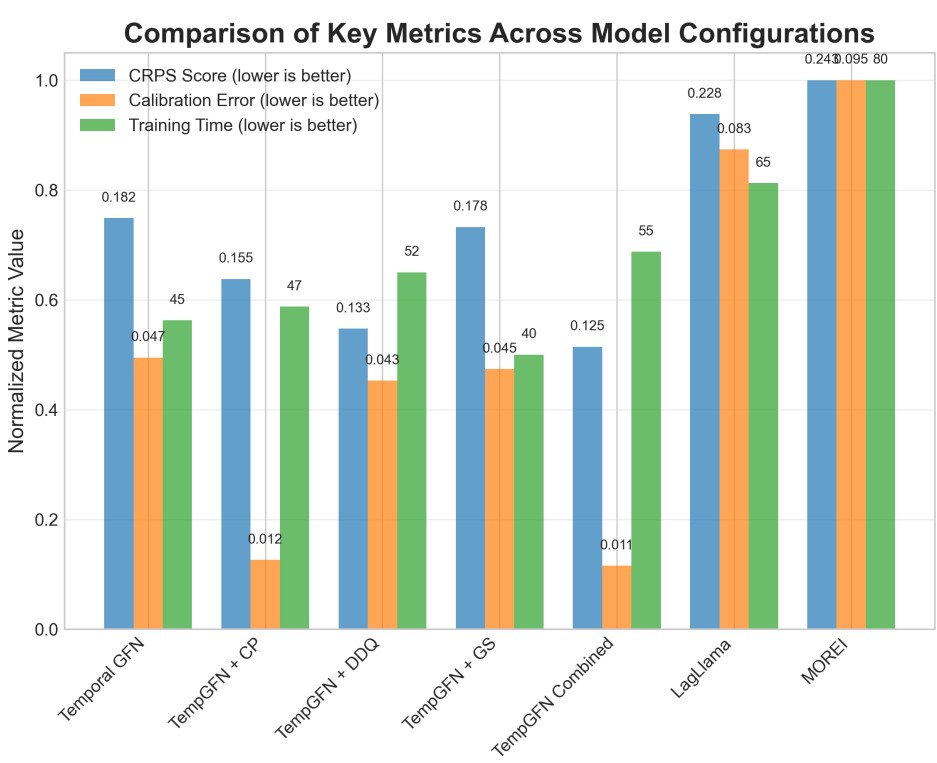

Figure 25: Comparison of key metrics across different model configurations.

# K   Broader Impacts

The development of Temporal Generative Flow Networks (Temporal GFNs) for probabilistic time series forecasting, while primarily an advancement in machine learning methodology, carries a spectrum of potential societal impacts that merit careful consideration. The most significant positive impacts stem from the potential to enhance decision-making in critical domains where understanding uncertainty is paramount. In healthcare, for instance, the ability of Temporal GFNs to provide more accurate and well-calibrated probabilistic forecasts for patient vital signs or physiological signals could lead to earlier detection of adverse events, optimized resource allocation in intensive care units, and more personalized treatment strategies. The framework's capacity to capture multimodality is particularly valuable here, as patients may exhibit distinct future health trajectories. Similar benefits extend to finance, where reliable probabilistic forecasts can improve risk management and portfolio optimization; to energy systems, where better predictions of demand and renewable generation can enhance grid stability and efficiency; and to climate science, where nuanced forecasts with uncertainty can better inform policy. Beyond specific applications, by offering a new way to learn distributions over complex sequential behaviors, Temporal GFNs might also contribute to a deeper scientific understanding of the underlying dynamics in diverse systems.

