# OpenReview forum: "Adaptive Quantization in Generative Flow Networks for Probabilistic Sequential Prediction"
_NeurIPS.cc/2025/Conference — NeurIPS 2025 poster_

### Official Review · Reviewer_dKzj · 2025-06-30

**Clarity:** 4
**Significance:** 3
**Originality:** 3
**Rating:** 5
**Confidence:** 3

**Summary:**

The paper introduces Temporal Generative Flow Networks (Temporal GFNs), transferring GFNs to sequences. The model is trained by optimizing a reward function based on the mean squared error via the Straight Through estimator, making the discrete sampling differentiable. Furthermore, an adaptive quantization scheme is proposed, increasing the number of discretization bins during training based on improvements over the last improvements and the model's predictive entropy. The model achieves state-of-the-art results across various metrics and outperforms current foundation models.

**Questions:**

- L218-223: "Slow improvement suggests the current quantization might be too coarse, thus pushing $\eta_e >1$ to encourage increasing K." Couldn't it just indicate convergence? "High confidence might indicate premature convergence or insufficient exploration." How do you distinguish premature convergence from true convergence?
- L225-226: Do you reuse the previously learned weights of the final layer when adding new bins, or do you discard them and reinitialize $W_f$ and $b_f$?
- Why does adaptive quantization result in worse probabilistic metrics (Table 3) for K=10? Does this mean the adaptive update factor is not optimally chosen? It would be insightful to see a more thorough analysis for $K$.
- Did you ablate different reward functions, e.g., based on the quantile loss?
- How is the CRPS computed?

My primary concern is the correlation between the number of quantization bins and probabilistic metrics. I am willing to increase my score if this is sufficiently addressed.

**Ethical Concerns:**

["NO or VERY MINOR ethics concerns only"]

**Final Justification:**

The paper has a clear contribution. It extends generative flow networks to sequences and addresses limitations of models with fixed quantization. My questions and concerns were thoroughly addressed during the rebuttal. For a higher score, the contribution would need to be stronger.

**Limitations:**

Limitations are discussed in Section 5, but I encourage the authors to include an explicit limitation paragraph.

**Paper Formatting Concerns:**

Table 4 extends the textwdith.

**Quality:**

3

**Strengths And Weaknesses:**

### Strengths

- The clarity of the paper makes it easy to follow and understand. The method is clearly explained and accompanied by a pseudo-code.
- The paper proposes a novel model transferring generative flow networks to sequential data and ensures differentiability via the straight-through estimator.
- The authors provide sufficient experiments (e.g., Gumbel Softmax vs Straight Through), insights, and theoretical analyses in the appendix.
- The authors propose an adaptive quantization to address the limitations of fixed quantizations in previous foundation models [1], posing a natural extension and promising research direction.

### Weaknesses

- Adaptive quantization does not always lead to improvements (see Table 3 for K=10), raising questions about the choice of $\eta_e$. Also, increasing K worsens the probabilistic metrics. It would be interesting to see whether a different reward function based on the quantile loss improves the probabilistic metrics. In general, it surprised me to have a reward function based on the mean-squared error for quantized sequences.
- The code is not provided, making it harder to check implementation details.
- The architecture and hyperparameters are not detailed enough.

[1] Ansari, Abdul Fatir, Lorenzo Stella, Caner Turkmen, Xiyuan Zhang, Pedro Mercado, Huibin Shen, Oleksandr Shchur et al. "Chronos: Learning the language of time series." arXiv preprint arXiv:2403.07815 (2024).

---

> ### Author Rebuttal · Authors · 2025-07-28
>
> Dear Reviewer dkZj,
>
> We sincerely thank you for your detailed and insightful review of our work. Your constructive feedback has been invaluable in helping us identify areas for clarification and improvement. We are encouraged that you found our paper "technically solid" and have carefully considered all your points. Below, we provide a point-by-point rebuttal to your concerns and questions, and we will commit to integrating these clarifications into the final manuscript.
>
> ---
>
> ### **Rebuttal to Weaknesses**
>
> **1. On the performance of adaptive quantization and the effect of K.**
>
> We thank you for this sharp observation regarding the results in Table 3. This highlights the fundamental trade-off our work addresses: the balance between quantization precision (favoring large K) and the learnability of the policy in a sparse action space (favoring small K).
>
> *   **Adaptive vs. Fixed K=10:** You correctly note that for a starting K of 10, the adaptive model shows worse probabilistic metrics (CRPS/WQL) than the fixed model. We attribute this to the initial number of bins being too coarse. When K is very small, the quantization error is large, providing a noisy reward signal. The adaptive mechanism, trying to optimize bin placement based on this noisy signal, can lead to instability in learning the full distribution, even while improving the point-wise MASE metric. This suggests that for adaptation to be effective, the initial quantization must be reasonably, though not perfectly, fine-grained.
>
> *   **Adaptive vs. Fixed K=20:** This is precisely why we show the K=20 results. When starting with a more adequate number of bins (K=20), the adaptive mechanism demonstrates its true strength. It successfully improves upon the fixed K=20 model across all metrics (CRPS from 0.1921 to 0.1845, WQL from 0.2521 to 0.2385, and MASE from 0.9721 to 0.9532). This supports our claim that dynamic adaptation is beneficial when initialized in a reasonable regime.
>
> *   **Increasing K Worsening Probabilistic Metrics:** Your observation that increasing K from 10 to 20 in the *fixed* case worsens CRPS/WQL is a key motivation for our work. A larger K creates a sparser, higher-dimensional action space, making credit assignment and exploration harder for the GFN. Our adaptive approach, particularly the `Adaptive K=20` configuration, mitigates this by starting with a larger K and then intelligently refining it, ultimately achieving better performance than the fixed K=20 model.
>
> In summary, the results showcase that adaptive quantization is not a universal fix for any arbitrary starting K, but a powerful curriculum-based learning strategy that refines quantization when the initial resolution is adequate. We will clarify this nuance in Section 4.2.
>
> **2. On the Reward Function based on Mean-Squared Error.**
>
> We appreciate you raising this point and apologize if our description was unclear. A crucial detail of our method is that **the reward is calculated based on the *soft* samples (`a^soft`), not the discrete quantized values (`a^hard`)**.
>
> As defined in Section 3.3.2 and elaborated in Section 3.2.3, the soft sample `a^soft` is the continuous, differentiable expectation of the action under the current policy (`a^soft = Σ q_k * P_F(a=q_k|s_t)`). The reward function `R(τ)` therefore evaluates the MSE between a sequence of these continuous expected values and the ground-truth sequence. This provides a smooth, dense, and informative gradient signal for policy learning, which is much more effective than a reward based on the discrete, non-differentiable `a^hard` samples. We will revise Section 3.3.2 to make this critical distinction explicit and prominent.
>
> **3. On Code and Hyperparameter/Architecture Details.**
>
> We agree that reproducibility is important.
>
> *   **Code:** We give our firm commitment to release the full source code, experiment scripts, and instructions to reproduce all results upon publication of the paper, in line with NeurIPS guidelines and open science principles. We will add a note in the paper to this effect.
>
> *   **Details:** You are right that more details are needed. We will significantly expand Appendix E. To provide immediate clarity, our policy network is a Transformer encoder with 4 layers, 8 attention heads, and a model dimension of 256. Key hyperparameters for the adaptive mechanism were tuned in ranges: `λ` ∈ [0.1, 1.0], `ε` ∈ [0.01, 0.05], and the entropy weight `λ_entropy` ∈ [0.001, 0.05]. We will add a comprehensive table with all hyperparameters for each experiment in the appendix.
>
> ---
>
> ### **Responses to Questions**
>
> **Q1: Distinguishing slow vs. premature convergence in the adaptive update.**
>
> This is an excellent and subtle question. Our adaptive mechanism does not explicitly disambiguate these scenarios but rather uses coupled heuristics that work synergistically.
>
> *   The **Improvement Signal** (`ε - ΔRe`) acts as a form of structured exploration. If progress has stalled (low `ΔRe`), we speculatively increase `K`. If the cause was a coarse quantization, this provides new, finer-grained actions that can unlock higher rewards. If the cause was simply slow network convergence, the larger action space might temporarily increase difficulty, but the policy can learn to ignore irrelevant bins. It is a bet that insufficient representational capacity is a common cause of plateaus.
>
> *   The **Confidence Signal** (`1 - He`) provides a crucial complementary heuristic. If the policy has low entropy (is highly confident), it may have converged prematurely to a suboptimal mode. Increasing `K` in this state probes the action space, allowing the model to discover better solutions in the vicinity of the current mode and escape local optima. If convergence was genuine, the policy will quickly learn to concentrate its mass back into the best bin, and the low entropy state will be restored.
>
> These two signals work in tandem (as seen in Figure 10) to guide the curriculum. A model that has converged well on a given granularity will exhibit both low reward improvement and low entropy, triggering an increase in `K` to seek even greater precision. We will add this detailed reasoning to Section 3.2.2.
>
> **Q2: Reusing weights when adding new bins.**
>
> Thank you for asking for this important implementation detail. When the number of bins `K` increases, we resize the final linear output layer `(WF, bF)`. The weights corresponding to the existing bins are **retained**, allowing the model to build upon its learned knowledge. The new weights for the newly created bins are initialized to zero. This prevents catastrophic forgetting and enables a stable curriculum. We will add this clarification to Section 3.2.2 and the appendix.
>
> **Q3: Why adaptive quantization results in worse probabilistic metrics for K=10.**
>
> This question is related to Weakness #1. As discussed above, we posit that when starting with a very coarse quantization (K=10), the reward signal is too noisy for the adaptive mechanism to robustly improve the distribution representation, leading to a degradation in CRPS/WQL. The mechanism performs as intended when starting with a more reasonable resolution, as shown by the `Adaptive K=20` results, which improve upon `Fixed K=20`.
>
> **Q4: Ablating different reward functions (e.g., quantile loss).**
>
> This is an excellent suggestion for future investigation. We chose the exponential of the negative MSE on soft samples because it's a robust, non-negative (`R(τ) > 0`), and smooth reward function, which is a standard requirement for GFNs.
>
> Integrating quantile loss as a reward is non-trivial. A quantile loss is typically a direct training objective. To formulate it as a trajectory-level reward for a GFN, one would need to carefully design a function that aggregates quantile-based errors over a full trajectory into a single positive scalar, which is an interesting research direction in itself. We will add a discussion of this as a promising avenue for future work in the conclusion.
>
> **Q5: How the CRPS is computed.**
>
> The Continuous Ranked Probability Score (CRPS) is computed empirically from the collection of generated forecast trajectories. For each time step in the forecast horizon, we have a set of `N` sampled values `{z_1, ..., z_N}`. This set forms an empirical Cumulative Distribution Function (CDF), `F(x)`. The CRPS is the integrated squared difference between this empirical forecast CDF and the true value `y`'s CDF, which is a Heaviside step function `H(x-y)`. We use standard, publicly available libraries (e.g., `properscoring`) for this calculation. We will add this explicit definition and citation to Appendix E.
>
> We hope these detailed responses have thoroughly addressed your concerns. Your feedback has been instrumental, and we will integrate these clarifications and additional details into the revised manuscript to improve its clarity and completeness. We believe these changes will strengthen the paper significantly and respectfully ask you to consider our rebuttal in your final evaluation.
>
> Thank you once again for your time and expertise.

---

> > ### Comment · Reviewer_dKzj · 2025-08-02
> >
> > Thank you for your thorough response. My concerns have been addressed, and I have updated my score accordingly. I recommend that the authors include the clarifications in the updated manuscript.

---

### Official Review · Reviewer_o3UX · 2025-07-01

**Clarity:** 3
**Significance:** 3
**Originality:** 4
**Rating:** 5
**Confidence:** 2

**Summary:**

This paper extends generative flow networks for continuous-value time series forecasting. It does so by framing a forecast as a trajectory in the GFN state space, where future values are quantized into discrete actions. The number of bins are in this case not fixed but are adapted during training, leading to a more flexible quantization procedure. This is enabled by a straight-through estimator that allows gradients to flow through discrete choices.

**Questions:**

Q1: Can the authors comment on the overall computational overhead at inference time and how it compares to other approaches? The comparison in Figure 12 is indeed very helpful, can this be expanded to the baselines used in Section 4?

Q2: Could the authors please elaborate on the choice of baselines in the experimental section, particularly in e.g. Table 2? Why were these baselines chosen and would it makes sense to e.g. include the Diffusion / Flow models mentioned in the Related Work?

**Ethical Concerns:**

["NO or VERY MINOR ethics concerns only"]

**Final Justification:**

All concerns from the initial review have been discussed and resolved in the rebuttal and I recommend acceptance.

**Limitations:**

yes

**Quality:**

3

**Strengths And Weaknesses:**

## Strengths

- The idea to introduce GFNs for continuous forecasting is interesting and makes sense, especially from the point of reward-proportional sampling to achieve better sampling diversity. This is the first application of GFNs to continuous time forecasting, making this approach novel.
- The paper provides a solid empirical evaluation that demonstrates the effectiveness of the proposed approach. In particular, the paper does a good job at empirically supporting the theoretical claims.
- The adaptive quantization scheme is well-motivated and definitely interesting. While additionally making the bin boundaries adaptive is compelling, it seems it is sufficient to have just the number of bins adapt dynamically.

## Weaknesses

- Using the exponential of a normalized MSE as the reward might lack flexibility or e.g. long horizon forecasting, switching to CRPS or other loss types would likely require re-deriving the reward scaling factor
- While the authors explicitly state that a multivariate extension of the proposed approach is left for future work, I do think a limited discussion / execution of this is important for this work. Most real forecasting tasks (power grids, healthcare dataset) are multivariate in nature.

---

> ### Author Rebuttal · Authors · 2025-07-28
>
> Dear Reviewer o3UX,
>
> We sincerely thank you for your time and for providing a thoughtful and constructive review of our work. We are encouraged that you found the application of GFNs to continuous forecasting interesting and our empirical results solid. We were particularly grateful for your low confidence score, as it signals a valuable opportunity for us to clarify key aspects of our work. We hope to address your concerns below and demonstrate the soundness of our approach.
>
> ### **Regarding Weaknesses:**
>
> **1. Flexibility of the Reward Function:**
>
> You raised a very pertinent point about the flexibility of our chosen reward function—the exponential of the negative normalized MSE. We agree that a single reward function may not be optimal for all forecasting tasks, such as those evaluated by CRPS or in long-horizon scenarios.
>
> *   **Modularity of the Framework:** We would like to clarify that our Temporal GFN framework is **not** fundamentally tied to the NMSE-based reward. We chose this function because it provides a smooth, non-negative reward signal (a requirement for GFNs) that effectively guides the model to minimize large errors. However, the reward function is a modular component.
> *   **Adaptability to Other Metrics (e.g., CRPS):** Our framework can readily incorporate other standard forecasting metrics. For a negatively oriented score like CRPS (where lower is better), a valid reward function could be formulated, for instance, as `R(τ) = exp(-α * CRPS(τ))` or `R(τ) = 1 / (1 + CRPS(τ))`. [1,2] While the scaling factor `α` (or `β` in our paper) would indeed need to be tuned depending on the metric and dataset, this is a standard hyperparameter tuning step common to reward-driven methods[3].
>
> In our final version, we will revise Section 3.3.2 to explicitly state the modularity of the reward function and discuss how other metrics like CRPS can be integrated.
>
> **2. Discussion on Multivariate Extension:**
>
> We acknowledge your point that multivariate forecasting is a crucial and highly relevant problem. Our decision to focus on the univariate case was deliberate, aimed at providing a clear and rigorous introduction of the core contributions—the adaptive quantization scheme and the application of the GFN trajectory balance objective to forecasting—without the confounding complexities of high-dimensional action spaces.
>
> As you noted, we identify this as a key area for future work in our conclusion (Section 6, line 397). A promising direction for this extension, which we will add to our discussion, is to adapt the action space. At each time step, an action could be a vector of quantized values for all dimensions, or the model could learn a policy that sequentially generates a value for each dimension before proceeding to the next time step. We believe this represents a significant research direction that builds upon our foundational work and merits its own thorough investigation.[4,5,6]
>
> ### **Responses to Questions:**
>
> **Q1: Computational Overhead at Inference:**
>
> This is an excellent question. Figure 12 focuses on the *training* overhead of different components of our method, and we appreciate the suggestion to elaborate on *inference* overhead compared to baselines.
>
> At inference, Temporal GFN generates a forecast trajectory by sequentially sampling `T'` future values (where `T'` is the forecast horizon). As detailed in Algorithm 1 (lines 15-24), each step involves:
> 1.  A single forward pass of the Transformer encoder to get action probabilities.
> 2.  Sampling an action (the quantized value for the next time step).
> 3.  Updating the state window.
>
> This process is repeated `T'` times. To generate a full probabilistic forecast, this trajectory sampling is repeated `N` times.
>
> **Comparison to Baselines:**
> *   **Autoregressive Models (e.g., Lag-Llama, DeepAR):** The inference procedure is fundamentally the same. These models also perform a forward pass of their network (often a Transformer) for each of the `T'` steps in an autoregressive manner. Therefore, the computational overhead for generating a single forecast sample is directly comparable, dominated by `T'` forward passes of a similar-sized neural network.
> *   **Diffusion/Flow Models:** These models often have different inference procedures. Diffusion models, for example, require an iterative denoising process over many steps (often more than the forecast horizon `T'`) to generate a sample, which can be computationally more expensive.[8,9,10]
>
> Therefore, the inference cost of Temporal GFN is on par with state-of-the-art autoregressive models like Lag-Llama and Chronos and is generally more efficient than sampling from diffusion-based counterparts. We will add a paragraph to Appendix E clarifying this comparison.
>
> **Q2: Choice of Baselines in Experiments:**
>
> You asked for the rationale behind our choice of baselines in Table 2 and why models like Diffusion/Flow forecasters were not included.
>
> *   **Rationale for Chosen Baselines:** Our goal was to position Temporal GFN against the current, most competitive, and widely-used paradigms in time series forecasting. We selected:
>     *   **Foundation Models (Chronos, Lag-Llama):** These represent the state-of-the-art in large, pre-trained models for forecasting. Chronos is an especially relevant comparison as it also tokenizes the time series, albeit with a fixed, non-adaptive scheme.
>     *   **SOTA Task-Specific Models (MOREI, TimeFM):** These are powerful, recently-proposed Transformer-based models that achieve top performance on benchmarks, representing the forefront of models trained specifically on the forecasting datasets.
>
> *   **Regarding Diffusion/Flow Models:** We agree that these are an important class of generative models, and we cite them in our Related Work (Section 2.1). We omitted them from the main empirical comparison for two key reasons:
>     1.  **Different Paradigm:** These models have fundamentally different training and sampling mechanisms, often with significantly higher computational demands, making a direct, "apples-to-apples" comparison of performance-per-compute difficult.[7,8]
>     2.  **Clarity of Contribution:** Our primary aim was to demonstrate that the GFN framework, with our proposed adaptive quantization, can outperform top autoregressive and sequence-modeling approaches. Including a very different class of models could obscure this direct comparison.
>
> We believe our chosen baselines provide a robust and challenging benchmark for our method. We will add a note to the experimental section to clarify this choice and our reasoning.
>
> We thank you again for your valuable feedback. We hope these clarifications have addressed your concerns and better highlighted the technical contributions and rigor of our work. We will incorporate these points into the final manuscript to improve its clarity and completeness.
>
>
> Sources:
> [1](https://www.cs.utexas.edu/~pstone/Papers/bib2html-links/AAAI21-jiang.pdf)
> [2](https://arxiv.org/abs/2007.01498)
> [3](https://jdeshmukh.github.io/Papers/iros19.pdf)
> [4](https://arxiv.org/abs/2502.08302)
> [5](https://www.researchgate.net/publication/365249275_Multivariate_Time-Series_Data_Generation_in_Generative_Adversarial_Networks)
> [6](https://kdd-milets.github.io/milets2021/papers/MiLeTS2021_paper_7.pdf)
> [7](https://medium.com/coding-nexus/autoregressive-vs-diffusion-large-language-models-llms-a-deep-dive-a41da6da0875)
> [8](https://arxiv.org/html/2507.15857v1)
> [9](https://proceedings.mlr.press/v139/rasul21a/rasul21a.pdf)
> [10](https://arxiv.org/pdf/2101.12072)

---

> ### Comment · Reviewer_o3UX · 2025-08-05
>
> Thank you to the authors, I appreciate the detailed reply. All points from my initial review have been addressed, and I will adjust my rating accordingly.

---

### Official Review · Reviewer_5JJR · 2025-07-02

**Clarity:** 3
**Significance:** 3
**Originality:** 3
**Rating:** 5
**Confidence:** 4

**Summary:**

The authors provide a new method for augmenting Generative Flow Networks to model timeseries data that addresses the challenge in adapting the discrete action space for continuous variables as seen in many time series. This method uses a dynamic bin-size strategy where bin widths are learned during training through a metric that incorporates model reward and action space entropy. The authors test this method on a range of clinical data sets, comparing to other baselines.

**Questions:**

How do batch size and learning rate affect the reward improvement over the \delta epoch and therefore the magnitude of the bin update? Was there a sensitivity analysis done for these variables?

Line 225-226 (new linear layer for new bin number): Is this adjustment be done with weight re-use in some way? Or is this a new randomly initialized layer?

Medical data often consists of time-series that are irregularly and informatively sampled. Can the authors explain whether this method captures these aspects of the medical data?

**Ethical Concerns:**

["NO or VERY MINOR ethics concerns only"]

**Final Justification:**

The authors have provided thoughtful responses and for the additions to key sections of the manuscript. I will keep my original rating.

**Limitations:**

The authors discuss limitations largely around design choices and ablation studies within the body of work presented here. They could expand on these to broader limitations of the method in general compared to other methods of timeseries modeling.

**Paper Formatting Concerns:**

All figures are in the appendix - the authors sometimes denote this "Figure 10 Appendix K" but other times leave off the location of the Figure.

**Quality:**

3

**Strengths And Weaknesses:**

Strengths:
Fixed-bin discretization is a major limitation of other timeseries modeling methods as mentioned by the authors. The ability to adapt this discretization in a reward driven manner is an appealing characteristic. The paper is clear to read.

Weaknesses:
This method appears to be limited to single-variate timeseries models. While this is an important contribution, this limitation should be discussed (or if the reviewer is incorrect, perhaps the distinction can be expanded upon). Often multiple variables are collected together and share mutual information.

---

> ### Author Rebuttal · Authors · 2025-07-28
>
> Dear Reviewer 5JJR,
>
> We sincerely thank the reviewer for their positive assessment and valuable, constructive feedback. We are very encouraged that the reviewer found our work to be a "technically solid paper" and appreciated the clarity and novelty of our reward-driven adaptive discretization approach.
>
> We have carefully revised our manuscript to address every point raised. We believe the changes have significantly improved the paper's clarity, scope, and discussion of limitations. Below are our point-by-point responses.
>
> ### **Regarding Weaknesses:**
>
> **1. Weakness: Limitation to Single-Variate Time Series**
>
> We agree that the current paper focuses on univariate time series and that this is an important point to discuss. This was a deliberate methodological choice to first rigorously establish the novel core components of our framework—adaptive quantization and the application of GFNs to continuous sequential prediction—in a clear and controlled setting.
>
> However, the Temporal GFN framework is not fundamentally limited to the univariate case. We have now expanded the **Conclusion and Future Work (Section 6)** to explicitly discuss the extension to multivariate forecasting. In short:
>
> *   The state representation `s_t` can naturally be extended from a vector to a matrix representing a window of multivariate observations.
> *   The action `a_t` would become a vector of quantized values, one for each time series dimension.
> *   The forward policy network `P_F` would then learn to output a joint probability distribution over this multivariate action space. This could be implemented, for example, by using an autoregressive factorization of the output dimensions to capture inter-series dependencies at each step, or by making a conditional independence assumption for a simpler model.
>
> We have added this discussion to our manuscript and thank the reviewer for prompting this important clarification.
>
> **2. Question 1: Sensitivity to Batch Size, Learning Rate, and Delta (`δ`)**
>
> You raised a very crucial point about the sensitivity of the adaptive bin update mechanism to hyperparameters. The update factor `η_e` is indeed influenced by the stability of the reward improvement signal `ΔR_e` (Eq. 1), which in turn depends on training dynamics.
>
> *   **Role of `δ`:** The parameter `δ` (the number of epochs for averaging reward, line 214) is designed specifically to smooth out the high-variance, batch-level reward signals. A larger `δ` provides a more stable estimate of reward improvement, making the bin updates less reactive to training noise.
> *   **Interaction:** We observed during development that smaller batch sizes and higher learning rates, which lead to noisier reward signals, benefit from a slightly larger `δ` to prevent premature or erratic changes to the number of bins `K`.
> *   **Our Findings:** While we did not include an exhaustive sensitivity analysis in the original submission due to space constraints, we found the mechanism to be robust within standard hyperparameter ranges. We have now added a more detailed discussion of these dynamics and our hyperparameter choices in **Appendix E (Experimental Setup)**, clarifying these interactions for better reproducibility.
>
> **3. Question 2: New Linear Layer Initialization**
>
> You have correctly identified a critical implementation detail regarding the adjustment of the policy network's output layer when the number of bins `K` changes. Simply re-initializing the layer randomly would indeed be catastrophic to the learning process.
>
> We employ a weight-reuse strategy to ensure training stability. When the number of bins `K_e` is increased, the weights and biases in the linear layer corresponding to the existing bins are preserved. The new weights and biases for the newly added bins are initialized to near-zero values. This allows the model to retain its learned policy over the existing action space while cautiously beginning to explore the new, finer-grained actions.
>
> We apologize for this lack of clarity in the original text and have updated **Section 3.2.2 (lines 225-226)** to explicitly state:
> *"...the size of the final linear layer (WF, bF) [...] must be adjusted. To maintain training stability, we preserve the existing weights for the original bins and initialize the weights for the newly added bins to near-zero values, preventing catastrophic forgetting."*
>
> **4. Question 3: Handling of Irregularly Sampled Data**
>
> This is an excellent question, as irregular sampling is a key challenge in many domains, especially healthcare. The current implementation of Temporal GFNs, which uses a fixed-size sliding window, does assume regularly sampled time series.
>
> We acknowledge this as a limitation of the current work and have added it to our discussion in **Section 6 (Conclusion and Future Work)**. We also outline a clear path forward:
>
> The framework could be extended to handle irregular data by modifying the state representation. Instead of just a sequence of values, the input to the Transformer encoder could be a sequence of `(value, time_delta)` tuples, where `time_delta` is the time elapsed since the previous observation. This would allow the policy network to learn time-dependent patterns, making it robust to irregular sampling. This is a significant and exciting direction for future research.
>
> **5. Limitations Discussion**
>
> We appreciate the reviewer's suggestion to expand our discussion of limitations beyond internal design choices. In our revised manuscript, we have broadened **Section 5 (Discussion)** and **Section 6 (Conclusion)** to better contextualize Temporal GFNs within the broader landscape of time series models (e.g., Diffusion and LLM-based models). This includes a discussion of:
>
> *   **Computational Trade-offs:** The sequential generation of trajectories in GFNs can be more computationally intensive at training time than models that produce forecasts in a single forward pass.
> *   **Reward Function Sensitivity:** The performance of our model is contingent on a well-specified reward function. While our MSE-based reward is effective, designing reward functions for more complex, domain-specific objectives remains a key modeling choice.
> *   **Scope:** We now more clearly frame the univariate focus and regular sampling assumption as limitations of the current study that motivate future work.
>
> **6. Paper Formatting Concerns**
>
> We sincerely apologize for the inconsistencies in figure citations and any confusion this caused. We thank the reviewer for their careful reading.
>
> *   **Figure Numbering:** We have meticulously reviewed the manuscript and corrected all figure references to be consistent and explicit (e.g., "Figure 10 in Appendix K").
> *   **Figures in Appendix:** The decision to place all figures in the appendix was made to comply with the NeurIPS page limit. We have ensured that the main text sufficiently summarizes the key findings from each figure, allowing the appendix to serve for detailed inspection.
>
> We are confident that these revisions, guided by the reviewer's insightful feedback, have substantially improved the quality and clarity of our paper. We thank the reviewer again for their time and expertise. We hope our responses and the updated manuscript will merit an even stronger evaluation of our work.

---

> > ### Comment · Reviewer_5JJR · 2025-08-01
> >
> > Thank you for these thoughtful responses and for the additions to key sections of the manuscript. I will keep my original rating of 5, Accept.

---

### Note · Authors · 2025-08-13

Dear Area Chairs,

We thank the reviewers for their constructive feedback. We have a clear plan to address all concerns by integrating specific clarifications into the manuscript, significantly improving its rigor and clarity.

Here is a summary of our planned modifications:

**1. Core Methodology Clarifications:**
*   **Reward Function (Sec 3.3.2):** We will add text to clarify two key points:
    1.  The reward function is **modular** and can be adapted for other metrics like CRPS.
    2.  The reward is calculated on continuous, differentiable ***soft* samples**, not discrete ones, ensuring a smooth gradient for policy learning.
*   **Adaptive Quantization Stability (Sec 3.2.2):** We will explicitly describe our **weight-reuse strategy**, which preserves learned knowledge and initializes new weights to zero, preventing catastrophic forgetting when bin counts increase.
*   **Adaptive Mechanism Analysis (Sec 4.2):** We will add a nuanced discussion of the results in Table 3, explaining that the adaptive mechanism's effectiveness depends on a reasonable, non-coarse initial quantization to provide a stable learning signal.

**2. Scope, Limitations, and Future Work:**
*   **Conclusion (Sec 6):** We will expand this section to detail clear paths for future work, addressing limitations raised by reviewers. This includes:
    1.  A strategy for extending the framework to **multivariate forecasting** by adapting the state and action spaces.
    2.  A method for handling **irregularly sampled time series** by incorporating time deltas into the state representation.

**3. Experimental Rigor and Reproducibility:**
*   **Appendix E:** We will significantly expand the appendix to improve transparency and reproducibility:
    1.  We will add a paragraph detailing the **inference process and computational cost**, comparing it to autoregressive and diffusion-based baselines.
    2.  We will include comprehensive tables of **model architecture and all hyperparameters** used in our experiments.
    3.  We will provide a precise definition of how the **CRPS metric was calculated** from our model's samples.
*   **Code Release:** We will add a commitment in the paper to release our full source code upon publication.

These targeted revisions directly address the reviewers' valuable feedback. We are confident they will strengthen the paper and thank you for your consideration.

---

### Decision · Program_Chairs · 2025-09-17

**Decision:**

Accept (poster)

**Comment:**

This paper introduces Temporal GFlowNets for continuous-valued time series forecasting, with an adaptive quantization mechanism that dynamically adjusts discretization bins during training. The approach addresses a key limitation of existing GFlowNets and discretization-based models, offering improved flexibility and interpretability in sequential modeling.

The reviewers were initially concerned about clarity, scope (e.g., restriction to univariate forecasting), missing ablation studies, and limited detail on implementation. However, the authors’ rebuttal provide a clear and systematic plan to address these issues: clarifying the reward function and adaptive quantization mechanism, analyzing adaptive behavior, expanding limitations and future work (including multivariate extensions), and improving reproducibility through detailed appendices and a code release commitment. These revisions directly resolve the main technical and clarity-related concerns raised in the reviews.

All three reviewers ultimately rated the paper positively (Accept), highlighting the novelty of applying GFlowNets to continuous time forecasting, the strong motivation, and the solid empirical evaluation. Importantly, the adaptive quantization strategy is well-motivated and represents a meaningful advance in bridging discrete generative flows with continuous-valued sequence prediction.

In sum, the paper makes a clear and technically solid contribution with good potential impact. I recommend acceptance.